

**Assimilation of Himawari-8 Imager Radiance Data with the WRF-3DVAR**
**system for the prediction of Typhoon Soulder**
**Dongmei Xu[1,2*],Aiqing Shu[1],Zhankui Zhang[1]**
*1. the Key Laboratory of Meteorological Disaster, Ministry of Education*
*(KLME)/Joint International Research Laboratory of Climate and Environment*
*Change (ILCEC)/Collaborative Innovation Center on Forecast and Evaluation of*
*Meteorological Disasters (CIC-FEMD), Nanjing University of Information*
*Science & Technology, Nanjing 210044, China*
*2.   Heavy Rain and Drought-Flood Disasters in Plateau and Basin Key Laboratory*
*of Sichuan Province, Chengdu, China*





## Abstract

Himawari-8 is a new generation geostationary meteorological satellite launched
by Japan Meteorological Agency (JMA). It carries the Advanced Himawari imager
(AHI) onboard, which can continuously monitor high-impact weather events with
high frequency space and time. The assimilation of AHI was implemented with the
framework of the mesoscale numerical model WRF and its three-dimensional
variational assimilation system (3DVAR) for the analysis and prediction of typhoon
"Soudelor" in the Pacific Typhoon season in 2015. The effective assimilation of AHI
Imager data in tropical cyclone with rapid intensify development has been
realized. The results show that after assimilating the AHI imager data under clear sky
conditions, the typhoon position in the background field in the model is effectively
corrected compared with the control experiment without AHI data. It is found that
assimilation of AHI imager data is able to improve the analyses of the water vapor
and wind in typhoon inner-core region. The analyses and forecast of the typhoon
minimum sea level pressure, the maximum near-surface wind speed, and the typhoon
track are further improved.
**Key words:** Weather Research and Forecasting model; Three-Dimensional
Variational Data Assimilation; AHI Imager Data; Typhoon



## 1. Introduction

In recent years, although researchers have made great progress in the field of NWP (numerical weather prediction), the huge challenges are encountered in the exact forecast of tropical cyclones (TCs) with quick intensifications (DeMaria et al., 2014). The predictability of these TCs is limited because it entails complex multi-scale dynamic interactions. These interactions include environmental airflows, TC vortex interactions, atmosphere-ocean interactions, and the effects of mesoscale and micro-convective scale, together with microphysics and atmospheric radiation. In order to attain a better initial condition (IC) and improve the accuracy of forecast, data assimilation seeks to fully utilize the observations. Most of TC's life span is over the ocean where conventional observations are relatively limited. Therefore, by analyzing observed data from the satellites and planes over the sea, it is crucial to adopt effective data assimilation (DA) methods to improve the analysis and forecast of TCs.

With the rapid development of atmospheric radiative transfer model (RTM), many numerical forecast centers now can adopt variational DA method to assimilate a variety of radiance data from different satellite observation instruments directly (Bauer et al., 2011; Buehner et al., 2016; Derber et al., 1998; Hilton et al., 2009; Kazumori et al., 2014; McNally et al., 2006; Prunet et al., 2000; Pennie, 2010). These data can take up 90% of all data used in global DA system and can improve NWP technique strikingly (Bauer et al., 2010). Some related researches demonstrated that in global model, satellite radiance DA makes more contributions to forecast accuracy



than conventional observation DA (Zapotocny et al., 2007).

Generally speaking, radiance data is derived from microwave and infrared

detecting instruments, which are from polar-orbit satellites and geostationary satellites,
respectively. Polar-orbit satellites cover the sphere of all the earth, so their
observations are suitable for global numerical forecast models (Jung et al., 2008).
Besides, compared to geostationary satellites, they have higher resolutions (Li et al.,
2017; Shen et al., 2015; Xu et al., 2013). However, it is highlighted that they are not
able to perform continuous observation over a fixed area, so this can leave out some
quickly intensified TCs or storms. On the contrary, because geostationary satellites
have a fixed location related to the earth's surface, although their resolutions are lower
than polar-orbit satellites, they can capture the formation and development of
mesoscale systems by continuous monitoring (Montmerle et al., 2007; Stengel et al.,
2009; Zou et al.,2011).

Geostationary satellites are able to continuously detect a region at a higher

frequency, thus supervising TCs over the vast ocean effectively. In fact, they can
capture convective spiral cloud systems relating to TCs and act as an important role in
TC's optimum observational position. As the first new generational geostationary
satellite, Himawari-8 was launched successfully in Sep 2014 by JMA (Japan
Meteorological Agency) and put into operation in July 2015 (Bessho et al., 2016). It
has an advanced imager called AHI (Advanced Himawari Imager) with 16 visible and
infrared bands, including 3 moisture channels, which can conduct a full-disk scan


every 10 minutes. Meanwhile, it can also acquire regional scan images and that is to
say it can scan the Japan and the target areas every 2.5 minutes. Compared to the early
geosynchronous imagers, AHI has more spectrum bands and this can monitor the state
of atmosphere with a higher frequency.
In recent years, some experts and scholars have carried out some researches on
geostationary satellite observation DA. Firstly utilizing GSI (Gridpoint Statistical
Interpolation) from NCEP (National Centers for Environmental Prediction), Zou, et al
(2011) conducted direct assimilation on imagers' data from GOES-11 and GOES-12
to estimate their potential influences on QPF (quantitative precipitation forecasts) of
coastal regions in the eastern part of American. They found assimilating radiance data
from GOES's imager has a remarkable improvement on 6 to 12 hour's QPF near
northern Mexico Gulf coast. Their work was continued by Qin, et al (2013), which
put thinned radiance data into GSI assimilation system to make a comprehensive
investigation on the issue on combined assimilation of GOES Imager data together
with AMSU-A (Advance Microwave Sounding Unit-A), AMSU-B (Advance
Microwave Sounding Unit-B), AIRS, MHS (Microwave Humidity Sounder), HIRS
(High Resolution Infrared Radiation Sounder), GSN (GOES Sounder). The results
showed the effect of single assimilation of AHI data is better than combined
assimilation in term of precipitation forecast. Zou, et al (2015) adopted GSI system to
assimilate radiance data from four infrared channels on GOES-13/15 and set up two
experiments for comparison. A symmetric vortex was used for initialization in the first



trial and an asymmetric counterpart for the other trial. Results showed that direct
assimilation of GOES-13/15's radiance data could generate continuous positive
effects on the track and intensity forecasts of tropical storm "Debbie" and this impact
was derived from assimilation of GOES radiance along with asymmetric vortex
initialization. Because himawari-8 has not been in operation for a long time, there are
few studies on himawari-8 data. Ma, et al (2017) used 4DEnVar (4D ensemble
variational data assimilation) in NCEP's GSI system to assimilate radiance of three
moisture channels of AHI under clear-sky condition and then NCEP GFS (Global
Forecast System) was utilized to estimate the impacts of AHI assimilation on whether
analysis and forecast. They found it had a positive influence on the forecast of global
vapor at high level of troposphere. Wang, et al (2018), based on 3DVAR system in
NWP center in northeast China operated by Liaoning Meteorological Bureau, firstly
attempted to conduct convective scale assimilation of AHI three moisture channels'
radiance data to study its impacts on the analysis and forecast of a rainstorm in
Northern China on 19th of Sep. It turned out that the assimilation of AHI radiance
could improve the simulated wind and vapor fields and the accuracy of rainfall
forecast in the first 6 hours obviously.
Although former researches have made several achievements in satellite data
assimilation and application, it is still a challenge to make more effective use of the
new generational geostationary satellite imager data with high spatial and temporal
resolution so that it can better satisfy the needs of meteorology. In most previous



studies, researches usually use a 6 hour's or even longer time interval with a coarse
spatial resolution. Therefore, until now hourly fast updating assimilation technique of
the stationary satellite radiance data in the convective scale in term of the analyses
and prediction of tropical cyclones has not been well carried out. This paper intends to
employ the new generational mesoscale WRF model and build an assimilation system
aimed at AHI imager data. Then a case of typhoon Soudelor is studied by performing
numerical simulation to address the impacts of convective assimilation on the
improvement of TC's IC and the enhancement of TC's track and intensity forecast.
**2. Observational data and DA system**
*2.1 An introduction to Himawari-8 AHI radiance data*
Himawari-8 satellite was launched by JMA (Japan Meteorological Agency) to a
geosynchronous orbit on 17 October 2014 and has begun its operational use since 7
July 2015. It is the first satellite of all new generational geosynchronous
meteorological satellites and plays a pioneering role for the geosynchronous imagers
to be launched in US, China, Korea and Europe. Himawari-8 is located between the
equator and 140.7°E, so the earth is observed between 60°N and 60°S meridionally
and between 80°E and 160°W zonally. Compared to its previous generation
Himawari-7, its detective ability can get remarkably improved since the instrument
AHI on Himawari-8. Besides, its device is comparable to imagers on American
GOES-R satellite (Goodman et al., 2012; Schmit et al., 2005; Schmit et al., 2008;
Schmit et al., 2017). AHI is able to provide a full-disk image every 10 minutes and





complete a scan over Japan every 2.5 minutes. AHI conducts continuous scan and
detection on a moving targeted typhoon. It has 16 channels covering visible,
near-infrared, and infrared spectral bands with a resolution of 0.5 km or 1 km, 0.5 km
or 1 km, and 2 km respectively. Channel 8 to 10 (6.2, 6.9, and 7.3 μm) are water vapor
bands that are sensitive to the humidity in the middle and upper troposphere (Di et al.,
2016). Other channels (channel 11, 12, 16: 8.6 μm, 9.6 μm, and 13.3μm ) are either
monitoring other fields such as the thin ice clouds, volcanic SO2 gas, the ozone or
CO2, or the atmospheric window channels (13-15: 10.4, 11.2, and 12.4 μm) function
as monitors for ice crystal/water, low water vapor, volcanic ash, SST (Sea Surface
Temperature) and other phenomena (Bessho et al., 2016).
Our work focuses mainly on assimilating the three moisture channels (6.2, 6.9,
and 7.3μm) since they are very sensitive to the humidity in the middle and upper
troposphere and have a certain effect on the lower troposphere. Thus, a large amount
of effective atmospheric information can be provided for AHI radiance data
assimilation in the troposphere.
*2.2 WRFDA system and AHI assimilation module*
WRFDA system is designed by National Center for Atmospheric Research
(NCAR) and it contains 3DVAR, 4DVAR, Hybrid parts. Our research is based on the
3DVAR method. An interface that is suitable for AHI DA is built in WRFDA system.
Currently, WRFDA is able to assimilate many conventional and unconventional
observation. In terms of satellite radiance observation, this system is compatible with
RTTOV (the Radiative Transfer model of the Television and Infrared Observational


Satellite (TIROS) Operational Vertical sounder) and CRTM (Community Radiative
Transfer Model) as observational operators. In this paper, CRTM is utilized as the
observational operator to simulate and compute AHI radiance data. Estimating the
systematic bias and random error of the observation data caused by the errors of
numerical models and instruments is the key to directly assimilate the satellite
radiance data. Apart from eliminating cloud pixels, other procedures to conduct
quality control are as follows. (1) when reading the data, remove the observed outliers
with the observed values below 50 K or above 550 K; (2) only the marine
observations are applied by removing the observation on the land and the more
complex observation points on the ocean surface; (3) remove observations when the
observation minus the background simulation is larger than 3 times of the observation
error; (4) the pixel point is removed when the CLW calculated by the background
field of the numerical model is greater than or equal to 0.2 kg/m2; (5) eliminate the
data when the observed value minus the background simulation value is greater than 5
K; (6) only vapor channels 8, 9, 10 on AHI are assimilated (Wang et al., 2018).
By using 3DVar algorithm, the assumption is that there is no bias between
observation and background (Dee et al., 2009; Liu et al., 2012; Zhu et al., 2014). A
bias correction scheme for observation is essential before DA. Usually, radiance bias
can be obtained by a linear combination of a set of forward operators.
$$\tilde{H}(x,\beta) = H(x) + \beta_0 + \sum_{i=1}^{N_p} \beta_i p_i \qquad (1)$$

Here, $H(x)$ represents the initial observation operator (before the bias
correction), x represents the mode state vector, $\beta_0$ represents a constant component
of the total bias (constant part), $p_i$ and $\beta_i$ represent the i-th predictor and its
coefficient respectively. In this study, four potentially state-dependent predictors



(1,000–300 hPa and 200–50 hPa layer thicknesses, surface skin temperature, and total
column water vapor) are applied. The variational bias correction (VarBC) scheme is
utilized to update the bias correction coefficient variationally with the new
observation operator considered in the cost function of 3DVar.

**3. Introduction to the case and experimental design**
*3.1 Typhoon Soudelor*
Typhoon Soudelor, that was happened in August, was the 13th typhoon in 2015
and became the second strongest tropical cyclone in this year. At 1200 UTC 30 July
2015, it formed at northwest Pacific Ocean as a tropical storm, located at 13.6° N,
159.2° E, then moved west by north. It upgraded to a strong tropical storm at 2100
UTC 1 August. Afterwards, it went through a process of rapid intensification. It
became a typhoon at 0900 UTC 2 August, a strong typhoon at 2100 UTC 2 August, a
super typhoon at 0900 UTC 3 August. Then it weakened to a strong typhoon in the
morning on August 5. However, it intensified to a super typhoon again at 1200 UTC 7
August with a maximum speed of 52 m/s, moving west by north, and its intensity
raised to its second peak. It was reduced to a strong typhoon again at 1800 UTC 7
August. It decreased to a typhoon, entering to Taiwan channel. It landed again as a
typhoon at 1410 UTC on the coast of Fujian province, China. Owing to continuous
orographic friction, it decreased to a tropical depression. Fig 1 shows the track of
Soudelor and different color lines represent typhoon's maximum wind speed. It is




displayed that after the formation of typhoon, its track is relatively stable. After July
30, its main body moved west by north at a speed of about 20 km/h. Its moving
tendency changed slightly within 10 days of its generation. However, its intensity
went through a rapid intensification, a weakening, a second intensification, then a
continuous weakening till disappearing gradually after landing Chinese mainland. Fig
2 demonstrates the variation of typhoon's intensity from July 31 to August 5. It is
shown that typhoon's maximum wind speed increased fast, while its minimum sea
level pressure decreased sharply. This was the stage of typhoon's rapid intensification.
We choose the date from August 1 to August 3 during its rapid intensification as our
research object.
*3.2 Experimental design*

Two experiments are designed to test the effects of AHI radiance data direct

assimilation on the analysis and forecast of Typhoon Soudelor starting from 1800
UTC 1 August to 0000 UTC 3 August. WRF 3.9.1 is employed as the forecast model
in our trial. We use Arakawa C grid in the horizon with a 5 km grid distance.
Vertically, it has 41 levels with 10 hPa as its top. Model center is (17.5 °N, 140 °E)
(Fig 4). Initial condition and lateral boundary are provided by 0.5°×0.5° GFS
reanalysis data. The following parameterization schemes are used: WDM6
microphysics scheme (Lim et al., 2010), Grell Devenyi cumulus parameterization
scheme (Grell et al., 2002), RRTM (Rapid Radiative Transfer Model) scheme
(Mlawer et al., 1997) and Dudhia scheme for longwave and shortwave radiation



respectively. Besides, YSU boundary layer scheme (Noh et al., 2003), Noah land
surface scheme are included.
The experimental procedures are illustrated by Fig. 3. Firstly, a 6 hour's spin-up
conducted at 1800 UTC 1 August before the forecast at 0000 UTC 2 August is used as
the background field for the assimilation. The first experiment is assimilating GTS
(Global Telecommunications System) conventional data (including aircraft report,
ship report, sounding report, satellite cloud wind data, ground station data) only,
which is called control experiment (CTNL). Another experiment is configured with
AHI radiance data assimilation (AHI_DA). AHI radiance data is assimilated hourly
further from 0000 UTC to 0600 UTC on August 2. Afterwards, an 18 hours forecast is
launched as the deterministic forecast. The climatological background error (BE)
statistics are estimated using the National Meteorological Center (NMC) method.
There are 5 control variables applied in this project including U component, V
component, full temperature (T), full surface pressure (Ps), and pseudo-relative
humidity (RHs). The observation error for each channel is estimated based on the O-B
from 0000 UTC on August 1, 2015 to 0000 UTC on August 3, 2015 every 6 hours.
Fig. 4 is the distribution of GTS observation data at the simulated domain at
0000 UTC 2 August. To avoid latent correlation among adjacent observation, we
choose 20 km to rarefy AHI observation data.
**4. Results**
*4.1 Minimization iterations*
Fig. 5 shows the change of cost function and gradient with the iteration times.
There is an obvious exponential decrease curve in Fig 5a, while Fig 5b shows gradient
decreases with the increase of iteration times. Taking Fig. 5a as an example, cost





function decreases very remarkably in the first 10 iterations. However, after 30 times
of iteration, the cost function curve becomes smooth gradually since only in the first
iteration, the differences between background field and observation are largest. With
continuous iterations, background field goes through continued adjustments. Finally,
the cost function tends to reach a stable minimum that represents the point when cost
function has its optimal solution. Besides, the gradient in Fig. 5b decreases stably as
the number of times of iteration. The exponential decline of the cost function and the
change trend of its gradient indicate that the assimilation effect is satisfying. The final
iterated analytical field is close to the observation.
*4.2 Analytical results of the brightness temperature*

Fig. 6 shows the distribution of observed brightness temperature, simulated

background brightness temperature, and simulated analytical field brightness
temperature of channel 8, 9, and 10 of AHI at 0000 UTC 2 August 2015. Fig. 6a is the
distribution of brightness temperature on channel 8 of AHI. The spiral cloud belt and
the eye area of Typhoon Soudelor are vividly shown with 49691 data counts. Fig. 6b
is a simulated distribution of background brightness temperature of AHI channel 8 by
model and it is generated by a 6 hours' deterministic prediction starting at 1800 UTC
1 August 2015. Although typhoon's spiral cloud belt and eye area are clear in the
background field, compared to observed distribution of brightness temperature, there
also exist some deviations. It can be seen from the background field and the typhoon
core area that the overall magnitude of the brightness temperature is higher than the



observation. This is mainly caused by a weaker simulated typhoon intensity in the
background than observation. Fig. 6c is the distribution of brightness temperature
after assimilating AHI radiance data. Spiral cloud belt structure of typhoon is clearly
displayed and the overall magnitude of the brightness temperature is similar to the
observation, indicating that assimilation of AHI radiance data can improve the
analysis of temperature and moisture remarkably. Fig. 6d, e, and f are the observed
brightness temperature of AHI channel 9, the simulated brightness temperature of
background field, and the simulated brightness temperature of analysis field,
respectively. We can find a similar phenomenon: compared to observation, a higher
background brightness temperature exists, while the simulated background brightness
by analytical field fits closer to the observation. Fig. 6g, h, i represent observational
brightness temperature, simulated background brightness temperature, and simulated
analytical brightness temperature on channel 10, respectively, and they have similar
effects as channel 8 and 9. Generally, the brightness temperature distribution of the
three channels is different mainly because the three channels have distinct absorptive
bands. From the spiral cloud belt region (orange) of the background field, obviously
the simulated background brightness temperature of three channels is higher than
corresponding observation, while after assimilating AHI radiance data, compared to
background brightness temperature, simulated analytical brightness temperature is
closer to the observation.

Fig. 6 shows the distribution of observed brightness temperature minus





background brightness temperature (OMB) and the observed brightness temperature
minus analytical brightness temperature (OMA) after the bias correction of AHI
radiance data from channel 8, 9, and 10 at 0000 UTC 2 August 2015. Fig. 6a is the
distribution of OMB brightness temperature after the bias correction. In the figure,
part of typhoon's spiral cloud belt is clearly visible. The brightness temperature in
typhoon's core area is low, while the brightness temperature in other areas is high.
The mean of observed OMB was -4.65 K, indicating that the background brightness
temperature is higher than the observation. Fig. 6b shows that the OMA value of most
pixels are below 0.02 K, indicating that the analytical field fitting the observation
after analyzing. It can be inferred from Fig. 6a, c, and e that the magnitude in OMB of
channel 10 is generally larger than that of channel 9, while that of the OMB of
channel 8 is the smallest. This is because the detection height of channel 10 is lower
than that of channel 8 and 9, which is most greatly affected by the cloud. Conversely,
the weight peak of the channel 8 is the highest, being the channel least affected by the
cloud. In general, the analytical brightness temperature match well with the observed
brightness temperature of all the three water vapor channels after the assimilation of
AHI radiance data.

Fig. 7 illustrates the effect of the bias correction for AHI radiance data at 0000

UTC 2 August 2015. Fig. 7a, d, g are the scatter plots of the observed brightness
temperature and the background brightness temperature field before the bias
correction. The abscissa represents the observed brightness temperature and the


ordinate represents the background brightness temperature simulated by CRTM
observation operator according to the mode background field. Fig. 7b, e, h are results
after bias correction, Fig. 7c, f, i are the scatter plots of observed brightness
temperature and analytical brightness temperature after bias correction. From Fig. 7a,
before the bias correction, the values from the observation and the background are
compariable, but most of the scatter points are below the diagonal line. This suggests
that the observed brightness temperature is higher than the background simulated
brighness temperature. From Fig. 7b, after the bias correction, observed warm bias is
corrected to some degree. From Fig. 7a, b, after the bias correction, the root mean
square error (RMSE) of OMB decreases from 1.864 K to 1.627 K, with the average
decreasing from 0.956 K to 0.358 K, proving the validity and rationality of the
variational bias correction. Compared to the result of Fig. 7b, the scatters in Fig. 7c
are more symmetrical, fitting closely to the diagonal line. The mean and RMSE were
also significantly reduced, suggesting that the analytical field is more similar to
observation than background field. Channel 9, 10 have a similar result, but with a
significantly reduced mean and RMSE, indicating that the background field and
analytical field of channel 9, 10 match better with the observation than channel 8 does.
Among them the RMSE of channel 10 reaches the minimum. In Fig. 7i, the RMSE of
channel 10 analytical field is only 0.234 K.
Fig. 8 shows the observation number, the mean, and the standard deviation of
OMB and OMA of assimilation channel 8, 9, and 10 before and after bias correction.




It can be seen from the figure that after quality control, 24057, 24181, 21785
observation data enter the assimilation system in channel 8, 9, and 10, respectively.
From the mean value of OMB before the bias correction, the value of the three
channels is relatively small, indicating that the simulated brightness temperature of
the three channels is close to the actual brightness temperature. The lowest mean of
0.3 K is found in channel 10, indicating that the simulated brightness temperature of
channel 10 is closest to the observed brightness temperature. Bias correction
effectively corrects the systematic bias and reduces the mean value of observation
residuals. After the bias correction, the OMB mean value of the three channels
significantly decreases to nearly 0 K. With the bias correction, the simulated
brightness temperature is almost the same as the observed brightness temperature. The
analysis of the standard deviation of OMB shows that the results are compariable
before and after the bias correction, indicating that the bias correction basically does
not change the spread of OMB. The standard deviation of OMA decreases by about
80% compared to OMB, indicating that the error distribution is greatly improved after
assimilation.
The RMSEs of the simulated brightness temperature by the model before
assimilation and assimilation against the observation is also calculated. Fig. 9 shows
the above RMSEs during the assimilation time for channels 8, 9, 10. As can be seen
from Fig. 9, RMSE decreases after each analysis of the AHI assimilation experiment
compared with the previous one. The most significant improvement is from the first




analysis moment of channel 8, where RMSE of the brightness temperature after
assimilation significantly decreases from 1.64 K to 0.46 K, possibly due to the largest
observation increment at the first analytical time. The one hour forecast after the
analysis basically makes brightness temperature of RMSE increase. Overall, the effect
of the analysis of the channel 10 is most significant.
*4.3 Analysis of the typhoon structure*
Fig. 10 shows the wind field at sea level and the distribution of water vapor at
0000 UTC 2 August 2015. The obvious cyclonic eddy circulation structures in the
core area of the typhoon are found in both fields, while the anti-cyclonic circulation
exists in the northwest quadrant of the typhoon. The mixing ratio of water vapor in the
region where the typhoon is located is very high and the wind field is cyclonic,
indicating that the typhoon has a continuous water vapor advection. This is conducive
to the enhancement of typhoon. According to the flow field of the control experiment
in Fig. 10a, it can be seen that the water vapor convergence in the center of the
typhoon region is weak with the low intensity, and the water vapor convergence zone
is small. As can be seen from Fig. 10b, after the assimilation of AHI radiance data, the
streamlines in the typhoon region become denser, indicating that the cyclonic
circulation is strengthened. Compared to the control experiment, the intensity and
distribution of the moisture convergence zone after the assimilation of AHI radiance
data are also more beneficial to the development of typhoon. This suggests that the
assimilation of AHI radiance data is able to significantly improve the large-scale



environmental field in the simulation region of the typhoon system.

*4.4 Track forecast*

In order to further evaluate the effect of AHI radiance assimilation, a 18-hour
deterministic forecast is launched at the end of two assimilation experiments. As can
be seen in Fig. 11a, at the beginning of the forecast, the initial location of the typhoon
of the two trials has a large bias. The location of the typhoon in the control experiment
has a relative east-southward bias, while the location of the typhoon in AHI_DA trial
is relatively close to the observation. During the following 6-hour forecast, the
typhoon track predicted by the CTNL continues moving west-south with the
environmental wind, while the track simulated by AHI_DA experiment match better
with the best track than that of the CTNL. In summary, the track of AHI_DA trial is
closest to the observation track during the entire 18-hour deterministic forecast. Fig.
11b is the typhoon track error predicted by the two experiments. At the initial time of
prediction, the track errors of CTNL and AHI_DA are significantly different, with
magnitude of 63.2km and 16.7km, respectively. During the subsequent 18-hour
forecast, the track error of the CTNL gradually increases with the forecast time
reaching 232.5km at the end of the forecast. In contrast, the track error of AHI_DA
experiment is better controlled within 95 km during the entire 18-hour deterministic
prediction process. In general, the average track error of the CTNL is 123.46 km, and
the average track error of AHI_DA experiment is 53 km, indicating a significant
improvement in the track prediction.

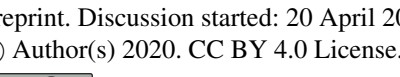


396  Fig. 12 discusses the time series of the typhoon intensity from the two

397 experiments with the maximum surface wind speed and minimum sea level pressure

398 (SLP) shown in Fig. 12a and Fig. 12b respectively. It can be seen that the maximum

399 near surface wind speed predicted by the CTNL is much lower than the actual wind

400 speed, mainly because the overall strength of Typhoon Soudelor simulated in the

401 background field of the model is relatively weaker. The maximum near surface wind

402 speed predicted by AHI_DA experiment fit closer to the best track with the maximum

403 difference about 2.6m /s after 12 hours forecast. In, Fig. 12b, the results of the

404 minimum SLP are consistent with Fig. 12a.

405 **5. Conclusion**

406  An interface for AHI data assimilation on the WRFDA system based on the

407 3DVAR assimilation method was built. Based on the Typhoon Soudelor in 2015, two

408 assimilation experiments for comparison was designed to examine the impact of AHI

409 moisture channel radiance data assimilation on the analysis and prediction of the rapid

410 development stage of typhoon under the condition of clear sky. Following conclusions

411 are obtained:

412 (1) The AHI imager on the new generation of geostationary meteorological satellite is

413 able to reflect the structure of Typhoon Soudelor very clearly. After a series of

414 pre-procedures such as the quality control, the bias correction, contaminated pixel

415 data is able to effectively be eliminated, ensuring the validity and rationality of the

416 observation data. The bias from the observations are also eliminated from the VarBC



statistical method, which is able to provide a positive impact on the data assimilation
procedure for the typhoon numerical simulation.
(2) Compared with the control experiment with the GTS data assimilation, the
3DVAR assimilation performed with AHI radiance data on top of the GTS data is able
to improve the structure of typhoon's core and outer rainband. Also, the position and
intensity of typhoon in the background field are able to be corrected.
(3) Compared to the predicted intensity and track of the control experiment and the
best track, it is found that the track, maximum wind speed, and minimum sea level
pressure from the AHI radiance data assimilation experiment are more similar to the
observation than the control experiment for the subsequent 18-hour forecast.

This paper realizes the AHI moisture channel radiance data assimilation under the

condition of clear sky. The results of the experiments indicate that AHI data
assimilation has a positive effect on the analysis and prediction of typhoon of the
rapid development stage of Typhoon Soudelor. Considering the complex influence of
underlying surface, only the rapid development stage of typhoon at sea were studied,
while the whole generation, development and disappearance stage of typhoon can also
be studied in the future. In addition, based on the AHI data of the water vapor
channels under the condition of clear sky, only 3DVAR method was adopted. Further
improvements under the condition of all sky and hybrid can be obtained in the future.



**Acknowledgments**


This research was primarily supported by the Chinese National Natural Science
Foundation of China (G41805016, G41805070), the Natural Science Foundation of
Jiangsu Province (BK20170940), the Chinese National Key R&D Program of China
(2018YFC1506404, 2018YFC1506603), the research project of Heavy Rain and
Drought-Flood Disasters in Plateau and Basin Key Laboratory of Sichuan Province in
China (SZKT201901, SZKT201904), and the Joint Open Project of KLME &
CIC-FEMD, NUIST (KLME201807, KLME201808).

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


**List of Figures**

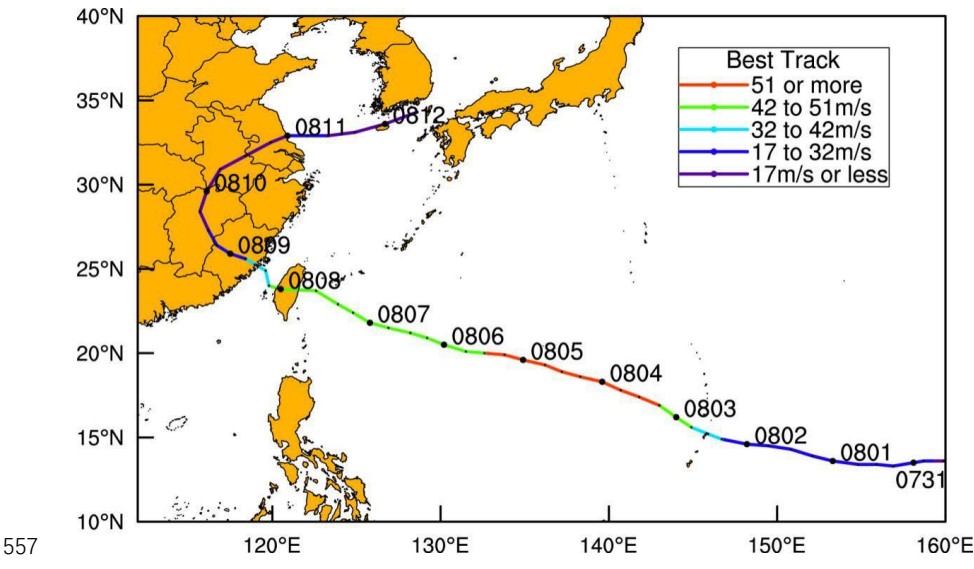


Fig.1 The track of Typhoon "Soudelor" in August 2015. Different colors represent
intensity changes.

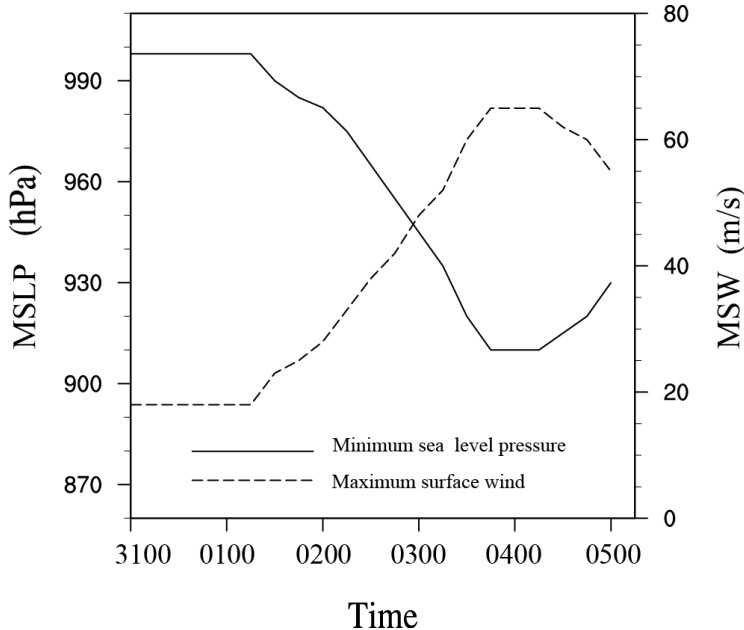





Fig.2 The time series of the minimum sea level pressure (solid line, unit: hPa) and the
maximum wind speed (dash line, unit: m/s) from July 31, 2015 to August 5, 2015.

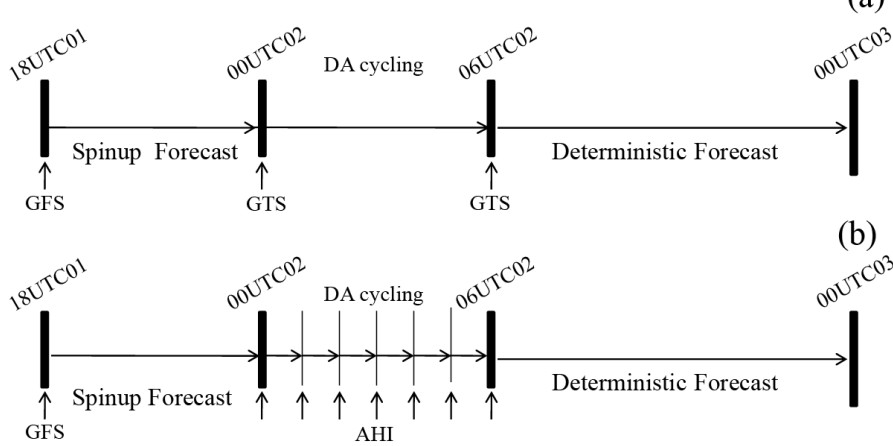

Fig. 3 The flow chart of experiments: (a) represents CTNL while (b) represents

AHI_DA

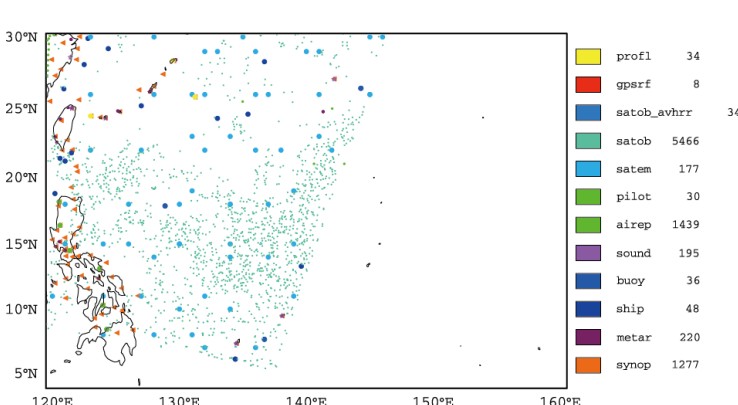

Fig. 4 Distribution of GTS in the simulated area at 0000 UTC 2 August 2015. On the
right side of the map is the name of observation data and the number of observations.

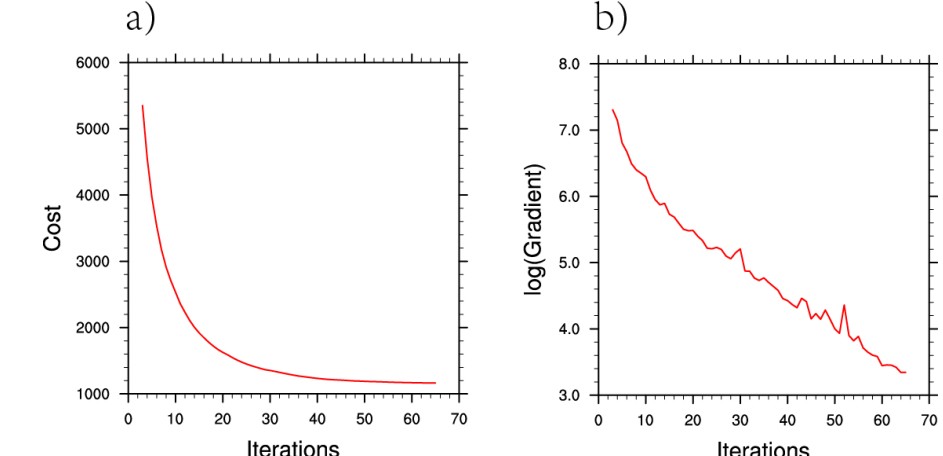

Fig. 5 (a) is a schematic diagram of the change of cost function with the number of

iterations, and (b) is a schematic diagram of the change of gradient with the number of

iterations.

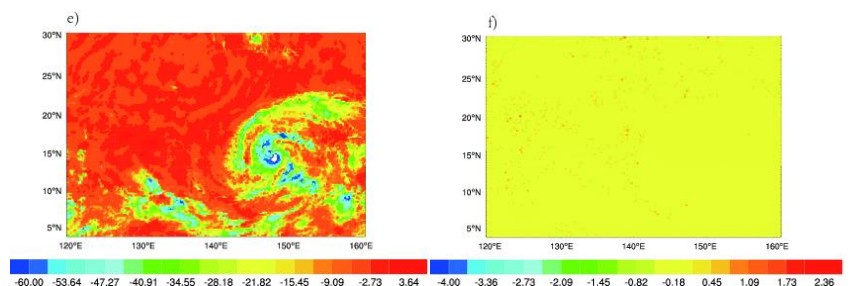



Fig. 6 (a, c, and e) represent OMB (unit: K) after bias correction for channel 8, 9, and
10, respectively; (b, d, and f) represent OMA (unit: K) after bias correction for

channel 8, 9, and 10, respectively at 0000 UTC 2 August 2015.



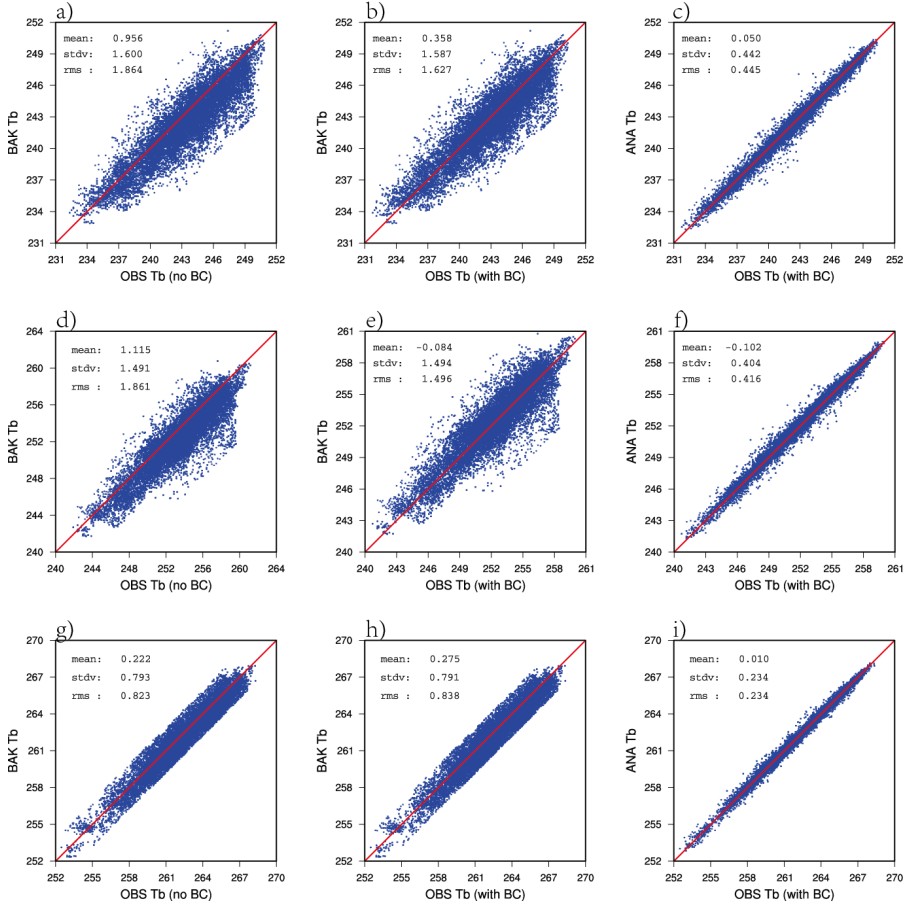


Fig. 7 Scatter plots of (a, d and g) the observed and background brightness

temperature before the bias correction of channel 8, 9 and 10. Scatter plots of (b, e

and h) the observed and background brightness temperature after the bias correction

of channel 8, 9 and 10. Scatter plots of (c, f and i) the observed and analyzed

brightness temperature after the bias correction of channel 8, 9 and 10.

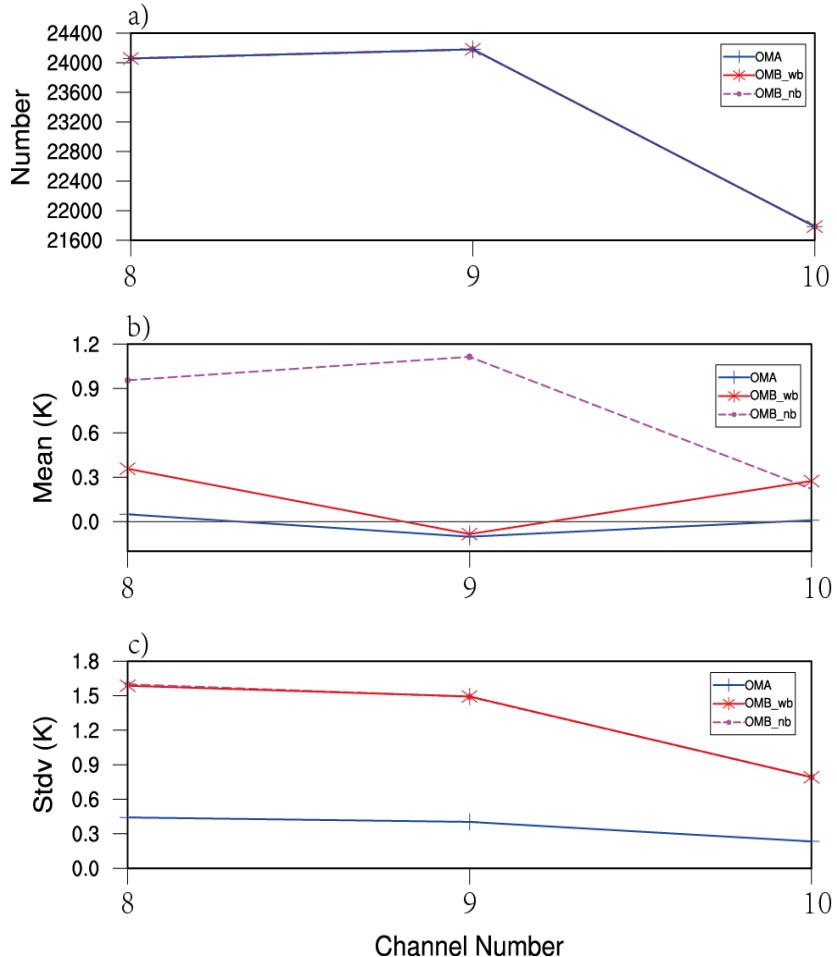


Fig. 8 Number of (a) observations, (b) mean (unit: K), and (c) standard deviations


(unit: K) of OMB and OMA before and after bias correction for water vapor channel
8-10 assimilation (OMB_nb: OMB without bias correction; OMB_wb: OMB with

bias correction).


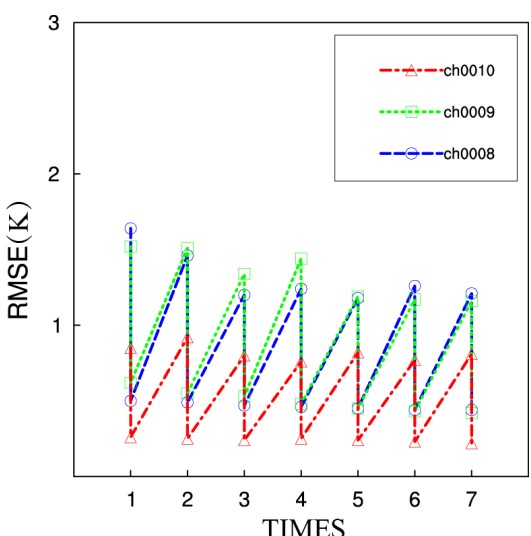


Fig. 9 Time series of the RMSE for the brightness temperature (unit: K) with

assimilation times before and after the data assimilation.

a)                                    b)

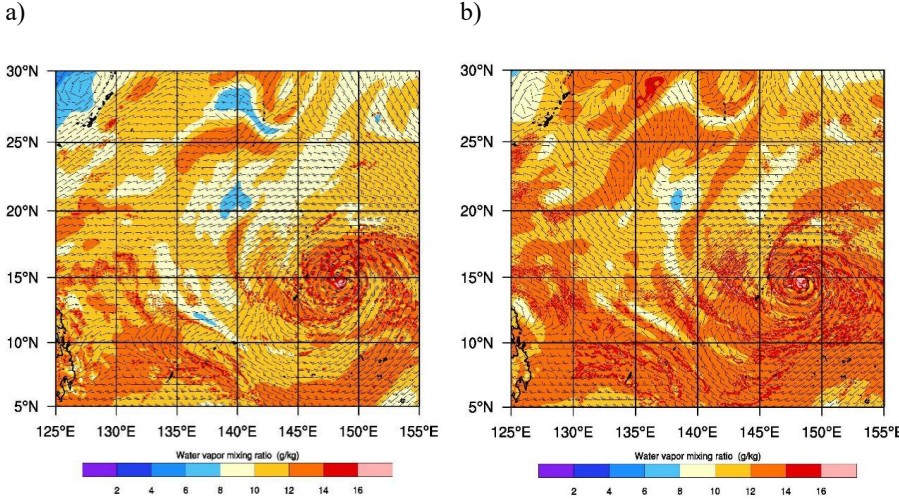

Fig. 10 The surface wind speed (vectors, unit: m/s) and water vapor (colored, unit:

597            g/kg) for (a) CTNL; (b) AHI_DA at 0000 UTC 2 August 2015.







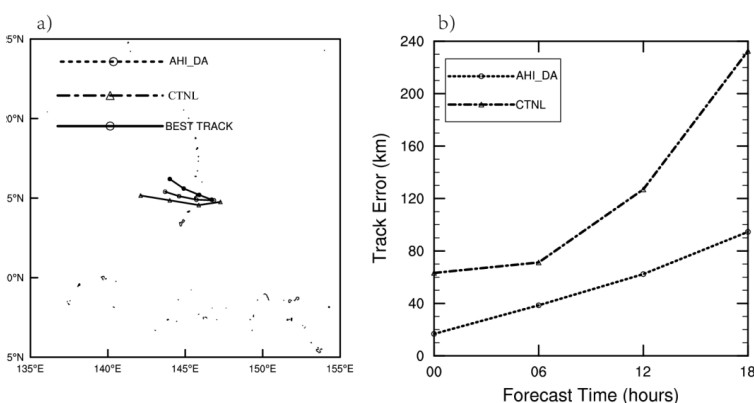

Fig. 11 The 18-hour (a) predicted tracks and the best track, (b) track errors (unit: m/s)

of Soulder from 0600 UTC 2 to 0000 UTC 3 August 2015.


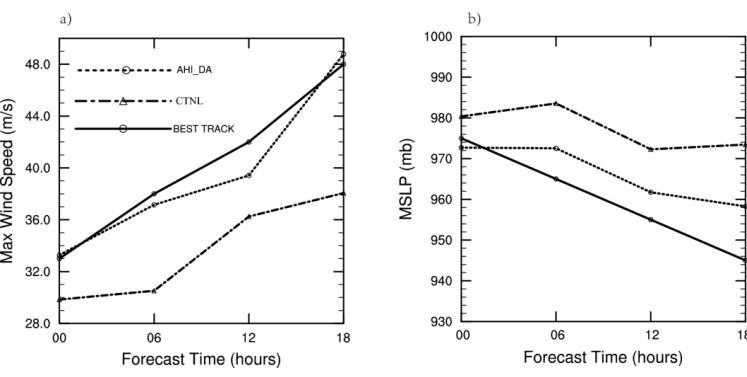

Fig.12 The 18-hour predicted (a) maximum surface wind speed (unit: m/s),(b)
minimum SLP (unit: hPa) of Soulder from 0600 UTC 2 to 0000 UTC 3 August 2015.