# Peer review of "Assimilation of Himawari-8 Imager Radiance Data with the WRF-3DVAR system for the prediction of Typhoon Soulder"

_Natural Hazards and Earth System Sciences, 2020_

## Referee Comment (RC1) · Anonymous Referee #1 · 3 Jun 2020

General comments, This study implemented the assimilation of JMA himawari-8 AHI radiance with the framework of the mesoscale numerical model WRF and its three-dimensional variational assimilation system (3DVAR) for the analysis and prediction of typhoon "Soudelor". The results are impressive in terms of the AHI radiance simulation and the forecast skill of the tropical cyclone for both the track and intensity. This action is meaningful, when geostationary meteorological satellite radiances provide valuable information of the weather systems with high spatial and temporal resolutions. However, there are several issues to be fixed to better clarify the methodologies and results. Specific comments,

1) section 4.1 Please explain why there is some oscillation in the variation of the gradient with increasing iteration? 2) This study assimilates clear-sky radiances. However, Figure 6 gives a very confusing picture. The data over the cloudy regions are presented in the observations. Radiances over the cloudy region are still calculated. Please provide the procedure for the verification results in Fig.6 with all the data shown as only clear-sky data are applied. 3) Please point out the reason channel 10 yield smaller RMSE?

Technical corrections 1) L35 use accurate instead of exact 2) L39 together with the microphysics and .. 3) L54 radiance data are .. 4) L104 positive impact 5) L106 Please reorganize the sentence "Wang, et al (2018)..."and check this kind of problem thoroughly for the whole manuscript 6) L112 previous researches 7) L147 This work focuses... Please check this problem for the whole manuscript 8) L243 Please change the word rarefy 9) L261 Fig. 6a shows or provides. Please fix this problem for the whole manuscript 10) L262 of channel 8 11) L332 after the bias correction. 12) L350 are also calculated 13) L427, this manuscript...
* * *

---

## Referee Comment (RC2) · Anonymous Referee #2 · 7 Jun 2020

This paper studies the effect of assimilating satellite observations on the prediction of typhoon. The predictions are made with WRF model, and initialization is performed by its 3D-VAR system. The technique is not new but the claim of novelty is that the system incorporates the newest data from a geostationary (in contrast to polar-orbiting) satellite, namely Himawari-8. Improvements in the predicted track and intensity of typhoon Soudelor are found with the assimilation of the satellite data. This is a timely study with potentially useful results. Nevertheless, clarifications are needed on some of the technical details:

(1) The conclusion of this work relies on a small number of runs without exploring

the dependence of the prediction on tunable parameters in WRF-3DVAR, for example those for the spatial correlation length and the scale of background variance. Previous studies have shown that the predictions of typhoon/hurricane tracks depend on those parameters (Xu et al. 2019, Meteorol. Appl., doi:10.1002/met.1820; Chou and Huang 2011, Adv. Meteorology, doi:10.1155/2011/803593). If this study just uses the default setting of those parameters, it would be useful to provide justifications or demonstrate that the results are robust with respect to tuning of the parameters.

(2) Since only clear-sky data is assimilated, one would guess that most of the satellite data over the cloudy area surrounding the core of typhoon are rejected. Yet, from Figs. 10(a) and 10(b) it appears that some distinctive small-scale structures (e.g., multiple spiral bands of high humidity) are created over the vortex core of the typhoon after the assimilation of satellite data. Are those structures artificial (e.g., due to numerical schemes of the model) rather than a realistic effect of assimilation of satellite observation? Related to this, it would also be interesting to compare the detail of the wind field near the center of the typhoon, but the vectors in Fig. 10 are hard to read. It would be useful to modify the figure to improve clarity.

(3) Figure 11(a), which shows the key result for typhoon tracks, is hard to read. The 3 tracks all look like solid lines that it is not possible to identify which is which. There seems to be random drawings in the background but it is not clear what they are (continental boundaries?) The labeling at left for the ordinate is cut off. Also, only one set of predictions is shown. What about other predictions made at different initial times? Do they exhibit similar behaviors? [This is also related to the comment in (1) concerning the robustness of results, given the small number of runs.]

---

## Referee Comment (RC3) · Anonymous Referee #3 · 7 Jun 2020

Overview: In this manuscript, the authors have tried to attempt to see the effect of data assimilation of AHI data for Typhoon Soudelor by using WRF-3DVAR. Perhaps, this data assimilation method has been widely used by many typhoon researchers, but this AHI data assimilation would be novel because Himawari-8 satellite provides more segmentalized bands compared to previous MTSAT series. I could see the effect of this AHI data assimilation, but the forecasting time is too short, which is close to nowcasting. Different initial locations may cause these track errors. Furthermore, the intensity forecasts from both experiments show the same tendency. It may indicate this AHI assimilation is not effective in improving inner-core structures yet. Actually, I could see longer period simulation results rather than 18-h simulations results to see definite

improvement of AHI_assimilation. Finally, as I suggest many specific comments and editorial comments, the authors seriously consider the English proofreading or carefully review this manuscript before publication. Therefore, since there are some corrections before publication, I would give a major revision opinion. Nevertheless, I think this paper approach is nice.

Specific comments

Lines 119-123: The authors could make the section lists to show how this paper is comprised.

Lines 69-72 & 126-128: These sentences are repeated.

Lines 147-151: I am not sure why the authors put the location of the focuses or purposes in this paragraph? I guess, if the authors put this to Introduction, it would be much clearer than now.

Lines 147-149: How can we understand that these three moisture channels are sensitive to those levels? Please provide some evidence.

Lines 190-200: About re-intensification of Soudelor, I guess, the authors reference JTWC best track data, because there is no such a pattern in JMA best track data. Please specify this information.

Line 200: what the Taiwan channel? Does that mean channel effect? I guess there are some papers discussing that. Please cite some references.

Lines 202-203: as mentioned in General comments, the authors should clarify the best track information.

Section 3.1: Figure 2 needs to make the same period with Fig. 1, and the authors may highlight the specific period (color-shading) according to the purpose.

Line 218: "We use Arakawa C grid in the horizon with a 5 km grid distance." What is the Arakawa C-grid? I know this grid-structure, but people reading this paper without

any background of the WRF model, may not understand this grid-type. If the authors want to use this, please clarify what it is or compare this with other grid-type kinds such as A, B, D, E types (should discuss momentum conservation and other kinds).

Line 219: is this eta levels? Or sigma levels? And is this even vertical spacing?

Figure 4 appears earlier than Fig. 3; that is not critical, but its order should be sequential. Please rephrase the sentences or remove them.

Lines 224-225: are there references for the Dudia scheme?

Line 225: It is surprising that YSU PBL is Noh et al. 2003? By the way, the authors said WRFV3.9.1. About the above and this line, the authors should carefully look at the WRF website to cite more appropriate references for the parameterizations.

Sections 3.1 and 4.4 and Fig. 1: What the authors reference the best track data? In the body context and Fig. 1, there is no information on that.

Section 4.2: I wonder that OMB and OBA indicate the observation (Himawari-8) – background (what background? Where it comes from? And how the authors calculate the brightness temperature of the analytic brightness temperature. Please clarify the methods of how to get the brightness temperature of the background and analytic one. Please put the title of each figure (band-8 micron unit).

Lines 300-307: I understand the authors' purpose. But these sentences should be more clarified. Without any vertical profile, it could be mere speculation. Please provide the weighting functions of each band, and it could then be discussed. And, the authors mentioned "cloud", it would be water vapors. In other words, most people may think "cloud" as "just cloud". As you know, there are many species such as ice, overwater phase, water vapor, and so on.

Figure 8: I wonder whether this stdv is statistically significant or not?

Figure 9: I could see the improvement; however, why they fluctuate?

Lines 363-366: It would be better if the authors cite one reference, at least for this sentence. Figure 11a: The legend is wrong. Please revise that. I guess, the figure could be enlarged.

Lines: 36-39: "The predictability of these TCs is limited because it entails complex multi-scale dynamic interactions. These interactions include environmental airflows, TC vortex interactions, atmosphere-ocean interactions, and the effects of mesoscale and micro-convective scale, together with microphysics and atmospheric radiation." Is your idea? Or someone said? Please cite some works for supporting this sentence.

Lines 241-243: Perhaps, this sentence accounts for data assimilation, in which the observations should be independent with each other. Do the authors think a 20-km resolution is appropriate to avoid the dependency between observations? If the authors say "right", please suggest any reference or results for that; for example, two normal-distributions.

Editorial comments

Over the whole manuscript: Please avoid to use many times "so" as the conjunction. Overall, the author should make the consistency of using the acronym and its order before the publication. For example, Lines 17-20: "The assimilation of AHI was implemented with the framework of the mesoscale numerical model WRF and its three-dimensional variational assimilation system (3DVAR) for the analysis and prediction of typhoon "Soudelor" in the Pacific Typhoon season in 2015.". Perhaps, the authors should correct some words; "AHI" → "AHI data"; spell "WRF" out; "typhoon Soudelor" → "Typhoon Soudelor (2015)". I guess the authors could use WRF-3DVAR "mesoscale numerical model weather research and forecasting three-dimensional variational assimilation system (WRF-3DVAR) or else. And please thoroughly see your wording to reduce mistyping or mistake since this paper goes forever after publication.

Lines 20-21 and else somewhere: "AHI Imager data" "AHI data" since the authors already used this as the acronym above. Please correct this word in the manuscript.

Line 21: ". . .rapid intensify. . ." Do the authors mean "rapidly intensifying"?

Lines 20-22: This line gives me something awkward. Do you mean "the AHI data assimilation was effective to simulate rapidly intensifying TCs."?

Line 26: forecast → forecasts

Line 35: Please use a general expression "quick intensification" → "rapid intensification"; "exact forecast" → "forecasts"

Line 40: The authors do not use the "IC" acronym after this line. Please remove this.

Line 42: ". . .relatively limited" → "relatively insufficient compared to the land" or proper expression.

Line 46: ". . . now can" → "have adopted"

Line 47: remove "directly"

Lines 50-51: "improve NWP technique" → I am not sure the authors want to say "these data improve NWP technique"? what the authors mean NWP technique? Please suggest examples.

Line 52: ". . . contributions to forecast accuracy . . ." → "contribution to improving the accuracy of the numerical model results", The authors should rephrase this sentence.

Line 58: "Besides, compared to geostationary satellites, they have higher resolutions (Li et al., 2017; Shen et al., 2015; Xu et al., 2013)." → "Besides, they have finer resolutions compared to geostationary satellites (Li et al., 2017; Shen et al., 2015; Xu et al., 2013).

Line 61: "quickly" → "rapidly"

Line 67: "supervising" → "observing"

Line 126: remove "(Japan Meteorological Agency)"

Lines 135: "satellite" → "satellite series"

Lines 190-191: The authors need to polish this sentence.

Line 193: "west by north" → "north-westwards"

Line 205: "its main body" → "tropical disturbance" or "tropical depression"

Line 220: "initial condition . . ." → "The initial condition and . . ."

Line 238-239: Please remove unnecessary acronyms such as "Ps", "RHs" which words are not used anymore.

Line 403: 2.6 m s-1

Please also note the supplement to this comment:
https://www.nat-hazards-earth-syst-sci-discuss.net/nhess-2020-120/nhess-2020-120-RC3-supplement.pdf

---

## Author Comment (AC1) · 21 Jul 2020

Reply to reviewer 1

General comments, This study implemented the assimilation of JMA himawari-8 AHI radiance with the framework of the mesoscale numerical model WRF and its three dimensional variational assimilation system (3DVAR) for the analysis and prediction of typhoon "Soudelor". The results are impressive in terms of the AHI radiance simulation and the forecast skill of the tropical cyclone for both the track and intensity. This action is meaningful, when geostationary meteorological satellite radiances provide valuable information of the weather systems with high spatial and temporal resolutions. How-
ever, there are several issues to be fixed to better clarify the methodologies and results. Specific comments, 1) section 4.1 Please explain why there is some oscillation in the variation of the gradient with increasing iteration? ————————————————
Reply:3DVAR works by minimizing the cost function through iterations, which will not guarantee the decrease for the gradient for each step monotonously. The similar oscillation in the gradient can be also found in Wang and Liu (2019). Reference: Wang, S. and Z. Liu, 2019: A radar reflectivity operator with ice-phase hydrometeors for variational data assimilation (version 1.0) and its evaluation with real radar data, Geosci. Model Dev., 12, 4031–4051.

2) This study assimilates clear-sky radiances. However, Figure 6 gives a very confusing picture. The data over the cloudy regions are presented in the observations. Radiances over the cloudy region are still calculated. Please provide the procedure for the verification results in Fig.6 with all the data shown as only clear-sky data are applied. ——————————————— Reply: The simulation of the brightness temperature is conducted as one of the verification methods. More explanation is added as "It should be pointed that even only parts of the AHI radiance data are applied after quality control in the data assimilation, the radiative transfer model is able to simulate the brightness temperature for all the pixels with the background and the analysis respectively for the verification purpose. The similar verification method is also applied in Yang et al., (2016)." Reference: Yang, C., Liu, Z., Bresch, J., Rizvi, S. R. H, Huang, X.-Y., and Min, J. AMSR2 all-sky radiance assimilation and its impact on the analysis and forecast of Hurricane Sandy with a limited-area data assimilation system, Tellus A: Dynamic Meteorology and Oceanography, 68, 1, 2016.

3) Please point out the reason channel 10 yield smaller RMSE? ——————————————— Reply: It is found from Fig. 8g that the simulated brightness temperature for assimilated pixels fit best with the observation compared to other two channels, which is likely related to strict cloud detection scheme for channel 10 with rather lower detecting peak. The authors have plotted the weighting function for the three water

vapor channels. Thus, the manuscript is revised as "Among them the RMSE of channel 10 is smallest as 0.234 K in Fig. 8i, which is likely related to strict cloud detection scheme for channel 10 with rather lower detecting peak (Wang et al., 2018)."

Technical corrections 1) L35 use accurate instead of exact ——————————————
Reply: Thanks for the helpful advice. Corrected at line 43

2) L39 together with the microphysics and .. ———————————————— Reply: Thanks. Added.

3) L54 radiance data are .. ———————————————— Reply: Thanks. Corrected.

4) L104 positive impact ———————————————— Reply: Thanks for pointing it out. Corrected.

5) L106 Please reorganize the sentence "Wang, et al (2018). . ."and check this kind of problem thoroughly for the whole manuscript ———————————————— Reply: Thanks for the helpful advice. The sentence is reorganized and the language is further edited by an English native speaker for the whole manuscript.

6) L112 previous researches ———————————————— Reply: Corrected.

7) L147 This work focuses. . . Please check this problem for the whole manuscript ——————————————— Reply: Corrected. We also improved these expressions with active words to an objective statement. Revisions can be found by tracks in detail.

8) L243 Please change the word rarefy ———————————————— Reply: Thanks for the helpful advice. We use thin to replace rarefy at line 283, now the sentence is "20 km is chosen to make thinning of AHI radiance data".

9) L261 Fig. 6a shows or provides. Please fix this problem for the whole manuscript ——————————————— Reply: Thanks. corrected.

10) L262 of channel 8 ———————————————— Reply: corrected.

11) L332 after the bias correction. ——————————————— Reply: Thanks. This expression is corrected as "before and after the bias correction", see line 353.

12) L350 are also calculated ——————————————— Reply: Thanks. This expression is corrected as "The RMSEs of the simulated brightness temperature by the NWP model before and after the assimilation are also calculated against the AHI radiance observations." at line 370.

13) L427, this manuscript. . . ——————————————— Reply: Thanks for the helpful advice. This expression is corrected as "In this study, the AHI radiance data assimilation is conducted under the clear sky condition." at line 460.

—————————————————

---

## Author Comment (AC2) · 21 Jul 2020

Reply to reviewer 2

This paper studies the effect of assimilating satellite observations on the prediction of typhoon. The predictions are made with WRF model, and initialization is performed by its 3D-VAR system. The technique is not new but the claim of novelty is that the system incorporates the newest data from a geostationary (in contrast to polar-orbiting) satellite, namely Himawari-8. Improvements in the predicted track and intensity of typhoon Soudelor are found with the assimilation of the satellite data. This is a timely study with potentially useful results. Nevertheless, clarifications are needed on some

of the technical details:

(1) The conclusion of this work relies on a small number of runs without exploring the dependence of the prediction on tunable parameters in WRF-3DVAR, for example those for the spatial correlation length and the scale of background variance. Previous studies have shown that the predictions of typhoon/hurricane tracks depend on those parameters (Xu et al. 2019, Meteorol. Appl., doi:10.1002/met.1820; Chou and Huang 2011, Adv. Meteorology, doi:10.1155/2011/803593). If this study just uses the default setting of those parameters, it would be useful to provide justifications or demonstrate that the results are robust with respect to tuning of the parameters. ——————
——————— Reply: Thanks for the pointing it out. The sentences are added as "The length scale and the variance scale are set to be 0.5 and 1 respectively after several sensitivity experiments conducted on tuning the background error. Similar conclusions are also found in Shen and Min (2015) with the scale factors related to the static background error covariance." to make it clear.

(2) Since only clear-sky data is assimilated, one would guess that most of the satellite data over the cloudy area surrounding the core of typhoon are rejected. Yet, from Figs. 10(a) and 10(b) it appears that some distinctive small-scale structures (e.g., multiple spiral bands of high humidity) are created over the vortex core of the typhoon after the assimilation of satellite data. Are those structures artificial (e.g., due to numerical schemes of the model) rather than a realistic effect of assimilation of satellite observation? Related to this, it would also be interesting to compare the detail of the wind field near the center of the typhoon, but the vectors in Fig. 10 are hard to read. It would be useful to modify the figure to improve clarity. ——————————— Reply: We agreed that most of the AHI clear-sky radiance data over the typhoon core area are rejected. However, the environment can be adjusted to some extent with the obtained observations. The model status in the cloudy area will also be modified with the spatial correlation in the background error covariance. The similar findings for small-scale information in the cloudy area can also be referred in Wang et al., (2018). Fig. 10 is

also replotted to improve the clarity. Related explanations are also added as "It should be pointed out that the model status in the cloudy area are modified due to the spatial correlation in the background error covariance. The similar findings for small-scale information in the cloudy area can also be referred in Wang et al., (2018)."

(3) Figure 11(a), which shows the key result for typhoon tracks, is hard to read. The 3 tracks all look like solid lines that it is not possible to identify which is which. There seems to be random drawings in the background but it is not clear what they are (continental boundaries?) The labeling at left for the ordinate is cut off. Also, only one set of predictions is shown. What about other predictions made at different initial times? Do they exhibit similar behaviors? [This is also related to the comment in (1) concerning the robustness of results, given the small number of runs.] ————————————
— Reply: Thanks for the helpful advice. We replotted the tracks with colorful lines in the revised manuscript. The random drawings in the background is also removed. The labeling at left is also kept. The forecast from 0000 UTC 02 August 2015 is also added for the track in Figure 12a. The forecast ranges are extended from 18 hours to 48 hours. In addition, the mean track errors, maximum surface wind speed error, and the minimum sea level pressure error are also calculated for two forecasts initialized at 0000 UTC and 0600 UTC presented in Figure 12c, and Figure 13.

---

## Author Comment (AC3) · 21 Jul 2020

Reply to reviewer 3

Overview: In this manuscript, the authors have tried to attempt to see the effect of data assimilation of AHI data for Typhoon Soudelor by using WRF-3DVAR. Perhaps, this data assimilation method has been widely used by many typhoon researchers, but this AHI data assimilation would be novel because Himawari-8 satellite provides more segmentalized bands compared to previous MTSAT series. I could see the effect of this AHI data assimilation, but the forecasting time is too short, which is close to nowcasting. Different initial locations may cause these track errors. Furthermore, the

intensity forecasts from both experiments show the same tendency. It may indicate this AHI assimilation is not effective in improving inner-core structures yet. Actually, I could see longer period simulation results rather than 18-h simulations results to see definite improvement of AHI assimilation. Finally, as I suggest many specific comments and editorial comments, the authors seriously consider the English proofreading or carefully review this manuscript before publication. Therefore, since there are some corrections before publication, I would give a major revision opinion. Nevertheless, I think this paper approach is nice.

Specific comments Lines 119-123: The authors could make the section lists to show how this paper is comprised. —————————————— Reply: Thanks for your advice. A new paragraph is added as "Section 2 describes the observations and the data assimilation system. Introductions to the typhoon case and the experimental setup are provided in section 3. The detailed results in terms of the analyses and the forecasts are illustrated in section 4 before conclusions are summarized in section 5."

Lines 69-72 & 126-128: These sentences are repeated. ——————————
Reply: Agreed. Related sentences are deleted in section 1 and section 2.1. The sentence is also reorganized as "As the first new generational geostationary satellite, Himawari-8 plays a pioneering role for the geosynchronous imagers to be launched in US, China, Korea and Europe." from line 84 to 86. In the second part, we revised as "Himawari-8 satellite was launched by JMA to a geosynchronous orbit on 17 October 2014 and has begun its operational use since 7 July 2015 (Bessho et al., 2016)." from line 153 to 155.

Lines 147-151: I am not sure why the authors put the location of the focuses or purposes in this paragraph? I guess, if the authors put this to Introduction, it would be much clearer than now. —————————————— Reply: Thanks for the helpful advice. Following the reviewer's suggestion, these sentences are moved to the 6th paragraph in the introduction part (line 141 to 145).

Lines 147-149: How can we understand that these three moisture channels are sensitive to those levels? Please provide some evidence. ———————————— Reply: Thanks. The evidence for the sensitive levels for three water vapor channels is provided as the weighing function in Fig. 1 in the revised manuscript (line 141 to 146). The sentence is revised as "Our study focuses mainly on assimilating the three water vapor channels (6.2, 6.9, and 7.3$\mu$m) since they are very sensitive to the humidity in the middle and upper troposphere and have a certain effect on the lower troposphere. Thus, a large amount of effective atmospheric information can be provided for AHI radiance data assimilation in the troposphere. The weighting functions for the three channels are provided in Fig. 1." in the manuscript.

Fig.1 Weighting function for Channel 8, 9, and 10.

Lines 190-200: About re-intensification of Soudelor, I guess, the authors reference JTWC best track data, because there is no such a pattern in JMA best track data. Please specify this information. ———————————— Reply: In this study, the best track data are provided by the China Meteorological Administration (Yu et al., 2007; Song et al., 2010). "Related information is added in section 3.1(line 215) and section 4.4 (line 407-408). Reference: Yu H, Hu C, Jiang L. 2007. Comparison of three tropical cyclone intensity datasets. Acta Meteorol. Sin. 21: 121–128. Song J-J, Wang Y, Wu L. 2010. Trend discrepancies among three best track data sets of western North Pacific tropical cyclones. J. Geophys. Res. 115: D12128, DOI: 10.1029/2009JD013058.

Line 200: what the Taiwan channel? Does that mean channel effect? I guess there are some papers discussing that. Please cite some references. ————————————
— Reply: Thanks for the helpful advice. Taiwan channel means Taiwan Strait, which is a 180-kilometer (110 mi)-wide strait separating Taiwan and mainland China. To avoid misunderstanding, we replace Taiwan Channel with Taiwan Strait at line 227.

Lines 202-203: as mentioned in General comments, the authors should clarify the

best track information. ————————————————— Reply: Agreed. The best track data are provided by the China Meteorological Administration (Yu et al., 2007; Song et al., 2010). "Related information is added in section 3.1(line 215) and section 4.4 (line 407-408).

Section 3.1: Figure 2 needs to make the same period with Fig. 1, and the authors may highlight the specific period (color-shading) according to the purpose. ————————————————— Reply: Agreed. Figure 2 is replotted from 0000 UTC 30 July 2015 to 0600 UTC 12 August 2015 and the specific period from 1800 UTC 1 August 2015 to 0000 UTC 3 August 2015.

Fig. 3 The time series of the minimum sea level pressure (solid line, unit: hPa) and the maximum surface wind (dash line, unit: m s-1) from 0000 UTC 30 July 2015 to 0600 UTC 12 August 2015.

Line 218: "We use Arakawa C grid in the horizon with a 5 km grid distance." What is the Arakawa C-grid? I know this grid-structure, but people reading this paper without any background of the WRF model, may not understand this grid-type. If the authors want to use this, please clarify what it is or compare this with other grid-type kinds such as A, B, D, E types (should discuss momentum conservation and other kinds). ————————————————— Reply: Agreed. Sentences are added to make it clear as "As is known, Arakawa A grid is "unstaggered" by evaluating all quantities at the same point on each grid cell. The "staggered" Arakawa B-grid separates the evaluation of the velocities at the grid center and masses at grid corners. Arakawa C grid further separates evaluation of vector quantities compared to the Arakawa B-grid." (lin246-250)

Line 219: is this eta levels? Or sigma levels? And is this even vertical spacing? ————————————————— Reply: Corrected. The eta levels are applied with coarser vertical spacing for the higher levels. The manuscript is revised as "Vertically, it has 41 eta levels using 10 hPa as its top with coarser vertical spacing for the higher levels."

(lin250-252) to make it clear.

Figure 4 appears earlier than Fig. 3; that is not critical, but its order should be sequential. Please rephrase the sentences or remove them. ————————————————
Reply: Agreed. Thanks for pointing it out. The order of Figure 3 and Figure 4 is changed in the revised manuscript and related sentences are rephased.

Lines 224-225: are there references for the Dudia scheme? ———————————
— Reply: Thanks for the helpful advice. The sentence is revised as "The following parameterization schemes are used: WDM6 microphysics scheme (Lim et al., 2010), Grell Devenyi cumulus parameterization scheme (Grell et al., 2002), RRTM (Rapid Radiative Transfer Model) longwave radiation scheme (Mlawer et al., 1997), shortwave radiation scheme (Dudhia et al., 1989), and YSU boundary layer scheme (Hong et al., 2006) ." now from line 254 to 262. Besides, the reference is added as follows, Dudhia, J. Numerical Study of Convection Observed during the Winter Monsoon Experiment Using a Mesoscale Two-Dimensional Model, Journal of the Atmospheric Sciences, 46, 3077-3107, 1989.

Line 225: It is surprising that YSU PBL is Noh et al. 2003? By the way, the authors said WRFV3.9.1. About the above and this line, the authors should carefully look at the WRF website to cite more appropriate references for the parameterizations. ———————————————————— Reply: Thanks. We double checked the details for all the physics from WRF user guide and make corrections for the reference of YSU PBL as, Hong S.Y., Noh Y., Dudhia J. A New Vertical Diffusion Package with an Explicit Treatment of Entrainment Processes. Mon. Wea. Rev., 134, 2318-2341, 2006. Follow the reviewer's suggestion, all the references for the for the parameterizations are also checked. WDM6: Lim, K.-S. S., and Hong, S.-Y.: Development of an effective double-moment cloud microphysics scheme with prognostic cloud condensation nuclei (CCN) for weather and climate models. Mon. Wea. Rev., 138, 1587-1612, 2010. Grell Devenyi cumulus parameterization: Grell G.A., Dévényi D.: A generalized approach to parameterizing convection combining ensemble and data assimilation techniques,

Geophys. Res. Let., 29, 587-590, 2002. The shortwave radiation scheme: Dudhia, J.: Numerical Study of Convection Observed during the Winter Monsoon Experiment Using a Mesoscale Two-Dimensional Model, Journal of the Atmospheric Sciences, 46, 3077-3107, 1989. The longwave radiation scheme: Mlawer E.J., Taubman S.J., Brown P.D., et al.: Radiative transfer for inhomogeneous atmospheres: RRTM, a validated correlated-k model for the longwave, Journal of Geophysical Research Atmospheres, 102: 16663-16682, 1997.

Sections 3.1 and 4.4 and Fig. 1: What the authors reference the best track data? In the body context and Fig. 1, there is no information on that. ——————————
—— Reply: Agreed. The best track data are provided by the China Meteorological Administration (Yu et al., 2007; Song et al., 2010). "Related information is added in section 3.1(line 215) and section 4.4 (line 407-408).

Section 4.2: I wonder that OMB and OBA indicate the observation (Himawari-8) – background (what background? Where it comes from? And how the authors calculate the brightness temperature of the analytic brightness temperature. Please clarify the methods of how to get the brightness temperature of the background and analytic one. Please put the title of each figure (band-8 micron unit). ——————————
Reply: The background for data assimilation is prepared as follows. Firstly, the initial condition and lateral boundary are obtained by the preprocessing module of WRF model with $0.5° \times 0.5°$ GFS reanalysis data. Then a 6-hour spin-up is conducted to provide as the background for the data assimilation purpose. The Community Radiative Transfer Model (CRTM; Liu and Weng, 2006) has been coupled within the WRFDA, which is applied as the observation operator for AHI radiance. The temperature and the humidity information from the model states are essential inputs for CRTM to calculate the simulated brightness temperature (the brightness temperature of the background and analysis). The simulation of the brightness temperature is conducted as one of the verification methods by comparing with the observed radiance. More explanation is added as "It should be pointed that even only parts of the AHI radiance data are

applied after quality control in the data assimilation, the radiative transfer model is able to simulate the brightness temperature for all the pixels with the background and the analysis respectively for the verification purpose. The similar verification method is also applied in Yang et al., (2016)." (line 309-313). Reference: Yang, C., Liu, Z., Bresch, J., Rizvi, S. R. H, Huang, X.-Y., and Min, J. AMSR2 all-sky radiance assimilation and its impact on the analysis and forecast of Hurricane Sandy with a limited-area data assimilation system, Tellus A: Dynamic Meteorology and Oceanography, 68, 1,2016. Liu, Q., and F. Weng, 2006: Advanced doubling-adding method for radiative transfer in planetary atmosphere. J. Atmos. Sci., 63(12), 3459–3465. We also have put the title of each figure with the band information along with the micron unit for the related figures.

Lines 300-307: I understand the authors' purpose. But these sentences should be more clarified. Without any vertical profile, it could be mere speculation. Please provide the weighting functions of each band, and it could then be discussed. And, the authors mentioned "cloud", it would be water vapors. In other words, most people may think "cloud" as "just cloud". As you know, there are many species such as ice, overwater phase, water vapor, and so on. ————————————— Reply: Agreed. The authors have plotted the weighting function for each channel in Fig. 1. Thus, the manuscript is revised as "It can be inferred from Fig. 7a, c, and e that the magnitude in OMB of channel 10 is generally larger than that of channel 9, while that of the OMB in channel 8 is the smallest. This is because the detection height of channel 10 is lower than that of channel 8 and 9 seen from the weighting function (Fig. 1), indicating channel 10 is largely affected by the clouds."

Fig.1 Weighting function for Channel 8, 9, and 10.

Figure 8: I wonder whether this stdv is statistically significant or not? ————— ————— Reply: Agreed. A significance test is conducted. A sentence is added as "The pairwise significance test is made between the OMA and OMB. Results show 95% confidence intervals in terms of the difference of the standard deviation using zero difference for the null hypothesis. A sentence is added as "Differences between

the standard deviations of the OMB and OMA were statistically significant at the 95% level using zero difference for the null hypothesis."

Figure 9: I could see the improvement; however, why they fluctuate? ——————————— Reply: Fig. 9 shows the RMSEs of the simulated brightness temperature by the model before and after data assimilation against the observations. The background before the assimilation is the short-term forecast from the previous analysis. The increase of the RMSE in the fluctuation arise from the model error in the short-term forecast. To make it clear, a sentence is added as "The background before the assimilation is the short-term forecast from the previous analysis. The increase of the RMSE in the fluctuation arise from the model error in the 1 hour short-term forecast." (line377-380).

Lines 363-366: It would be better if the authors cite one reference, at least for this sentence. ———————————— Reply: Agreed. One reference is added to show the correlation between the water vapor environment and the typhoon intensity as follows, Kamineni, R., et al., 2003: Impact of High Resolution Water Vapor Cross-Sectional Data on Hurricane Forecasting, Geophysical Research Letters, 30, 38-1.

Figure 11a: The legend is wrong. Please revise that. I guess, the figure could be enlarged. ———————————— Reply: Thanks for the helpful advice. We replotted the tracks with colorful lines in the revised manuscript. The random drawings in the background is also removed. The labeling at left is also kept. The forecast from 0000 UTC 02 August 2015 is also added for the track in Figure 12a. The forecast ranges are extended from 18 hours to 48 hours. In addition, the mean track errors, maximum surface wind speed error, and the minimum sea level pressure error are also calculated in Figure 12b, and Figure 13.

Lines: 36-39: "The predictability of these TCs is limited because it entails complex multi-scale dynamic interactions. These interactions include environmental airflows, TC vortex interactions, atmosphere-ocean interactions, and the effects of mesoscale

and micro-convective scale, together with microphysics and atmospheric radiation."
Is your idea? Or someone said? Please cite some works for supporting this sentence. ————————————— Reply: Corrected. The following reference is added, which describes the complex multi-scale dynamic interactions for the TCs. Reference: Minamide, M., and F. Zhang, 2018: Assimilation of all-sky infrared radiances from himawari-8 and impacts of moisture and hydrometer initialization on convection-permitting tropical cyclone prediction. Mon. Wea. Rev., 146, 3241–3258.

Lines 241-243: Perhaps, this sentence accounts for data assimilation, in which the observations should be independent with each other. Do the authors think a 20-km resolution is appropriate to avoid the dependency between observations? If the authors say "right", please suggest any reference or results for that; for example, two normal distributions. ————————————— Reply: Agreed. It is proved that raw radiance observations thinned to a grid with 2–6 times of the model grid resolution are able to remove the potential error correlations between adjacent observations (Schwartz et al ., 2012; Xu et al ., 2015; Choi et al., 2017). Also, sensitivity experiments with 25 km, and 30 km thinning mesh are also conducted with similar results. Thus, the manuscript is revised as "It is proved that raw radiance observations thinned to a grid with 2–6 times of the model grid resolution are able to remove the potential error correlations between adjacent observations (Schwartz et al ., 2012; Xu et al ., 2015; Choi et al., 2017). Hence, 20 km is chosen to make thinning of AHI radiance data. Also, sensitivity experiments with 25 km, and 30 km thinning mesh are also conducted with similar results." to make it clear.

Editorial comments Over the whole manuscript: Please avoid to use many times "so" as the conjunction. Overall, the author should make the consistency of using the acronym and its order before the publication. For example, Lines 17-20: "The assimilation of AHI was implemented with the framework of the mesoscale numerical model WRF and its three dimensional variational assimilation system (3DVAR) for the analysis and prediction of typhoon "Soudelor" in the Pacific Typhoon season in 2015.". Perhaps,

the authors should correct some words; "AHI" ! "AHI data"; spell "WRF" out; "typhoon Soudelor" ¡'Typhoon Soudelor (2015)". I guess the authors could use WRF-3DVAR "mesoscale numerical model weather research and forecasting three-dimensional variational assimilation system (WRF-3DVAR) or else. And please thoroughly see your wording to reduce mistyping or mistake since this paper goes forever after publication. ———————————————— Reply: Thanks. We change "so" to other conjunctions at line 68, 73, 132, 158. Also, consistency is considered and we change the sentence from line 23 to 27. Other revisions can be found by tracks in detail.

Lines 20-21 and else somewhere: "AHI Imager data" "AHI data" since the authors already used this as the acronym above. Please correct this word in the manuscript. ———————————————— Reply: Agreed. For an accurate expression, we use "AHI radiance data" in the whole manuscript.

Line 21: ". . .rapid intensify. . ." Do the authors mean "rapidly intensifying"? ————————————————— Reply: Corrected. We revised it as "The effective assimilation of AHI radiance data in improving the forecast of the tropical cyclone during its rapid intensification has been realized." at line 27-30.

Lines 20-22: This line gives me something awkward. Do you mean "the AHI data assimilation was effective to simulate rapidly intensifying TCs."? ——————————————— Reply: Corrected. This is what we mean. To avoid misunderstanding, the expression is changed to "The effective assimilation of AHI radiance data in improving the forecast of the tropical cyclone during its rapid intensification has been realized." at line 27-30.

Line 26: forecast ! forecasts ————————————— Reply: Corrected. "forecast" is replaced with "forecasts".

Line 35: Please use a general expression "quick intensification" ! "rapid intensification"; "exact forecast" ! "forecasts" ————————————— Reply: Thanks. The corresponding parts are corrected for the whole manuscript.

Line 40: The authors do not use the "IC" acronym after this line. Please remove this. —————————————— Reply: Thanks. "IC" acronym is removed at line 49.

Line 42: ". . .relatively limited" ! "relatively insufficient compared to the land" or proper expression. —————————————— Reply: Corrected. "relatively insufficient compared to the land" is used at line 51-52.

Line 46: ". . . now can" ! "have adopted" —————————————— Reply: Thanks for pointing it out. "have adopted" is used at line 56.

Line 47: remove "directly" —————————————— Reply: Thanks. "directly" is removed.

Lines 50-51: "improve NWP technique" ! I am not sure the authors want to say "these data improve NWP technique"? what the authors mean NWP technique? Please suggest examples. —————————————— Reply: Thanks for the helpful advice. Here we want to express abundant satellite data are crucial to the improvement of NWP accuracy because most part of the earth is covered by ocean where conventional observations are scarce. To avoid misunderstanding, "improve NWP technique" is replaced by "improve the accuracy of the numerical model results" at line 61-62.

Line 52: ". . . contributions to forecast accuracy . . ." ! "contribution to improving the accuracy of the numerical model results", The authors should rephrase this sentence. —————————————— Reply: Thanks. The sentence is rephrased at line 63-64.

Line 58: "Besides, compared to geostationary satellites, they have higher resolutions (Li et al., 2017; Shen et al., 2015; Xu et al., 2013)." ! "Besides, they have finer resolutions compared to geostationary satellites (Li et al., 2017; Shen et al., 2015; Xu et al., 2013). —————————————— Reply: Thanks for the helpful advice. The sentence is revised at line 69-72.

Line 61: "quickly" ! "rapidly" —————————————— Reply: Thanks. "quickly" is replaced with "rapidly" at line 74.

Line 67: "supervising" ! "observing" ——————————————— Reply: Thanks. "supervising" is replaced with "observing" at line 80.

Line 126: remove "(Japan Meteorological Agency)" ——————————— Reply: Corrected. "(Japan Meteorological Agency)" is removed at line 153.

Lines 135: "satellite" ! "satellite series" ————————————— Reply: Thanks for the helpful advice. This sentence is repeated and we delete it. From line 155 to 157.

Lines 190-191: The authors need to polish this sentence. ————————————
Reply: Thanks for the helpful advice. The sentence is revised as "From the record of the China Meteorological Administration (CMA), Typhoon Soudelor was the 13th typhoon in 2015 as the second strongest tropical cyclone in that year." at line 215.

Line 193: "west by north" ! "north-westwards" ——————————— Reply: Thanks for the helpful advice. We use "north-westwards" instead of "west by north" at line 219.

Line 205: "its main body" ! "tropical disturbance" or "tropical depression" —————————
——————————— Reply: Thanks. "its main body" is substitute with "the tropical depression" at line 231.

Line 220: "initial condition . . ." ! "The initial condition and . . ." ———————————
——— Reply: Thanks for pointing it out. "The initial condition and . . ." is used at line 252.

Line 238-239: Please remove unnecessary acronyms such as "Ps", "RHs" which words are not used anymore. ————————————— Reply: Thanks. These unnecessary acronyms are removed at line 274-275.

Line 403: 2.6 m s-1 ——————————————— Reply: Thanks for the helpful advice. "2.6 m s-1" is used at line 437.

———————————————————

2020-120, 2020.

**Fig. 1.** Fig.1 Weighting function for Channel 8, 9, and 10.

**Fig. 2.** Fig. 3 The time series of the minimum sea level pressure (solid line, unit: hPa) and the maximum surface wind (dash line, unit: m s-1) from 0000 UTC 30 July 2015 to 0600 UTC 12 August 2015.

---

## Referee Report (RR1)

**Assimilation of Himawari-8 Imager Radiance Data with the WRF-3DVAR 2 system for the prediction of Typhoon Soulder**

**Overview**

This manuscript has been significantly improved from the original manuscript. Nevertheless, I have found some mistakes. So, I want to request a revision to the authors. In particular, please go through deep consideration for the following comments.

**Major comment**

1. **A very critical error in title and Fig. 13. "Typhoon Soulder" →"Typhoon Soudelor",** Also, please specify Soudelor's year for the clarity.

2. *Line 232:* "Grell Devenyi cumulus parameterization scheme (Grell et al., 2002)," By the way, it is not always true. Nevertheless, since the authors set a 5-km spatial resolution in this study, I am very wondering why the authors did activate Grell Devenyi cumulus parameterization?

**Minor comments**

1. Lines 59-62: the sentence starts with "some researches …" but the authors cited one paper" Please cite more papers or revise this sentence.

2. Lines 68-69: I cannot understand what the authors mean (in bold) "it is highlighted that they are not able to perform continuous monitoring over a fixed area, **thus leaving out some rapidly intensified TCs or storms.**"

3. Line 70: "because geostationary satellites have a fixed location related to the earth's surface," it could potentially give a misunderstanding to the reader. Please just say "rotate with the earth".

4. Line 76: "In fact, they can capture convective spiral cloud systems relating to TCs." Since the geostationary satellites can capture more features related to TCs, the authors need to consider make a list or remove this sentence.

5. Lines:236-262: is there any reason why the authors explain first Figs. 5 and 6 and followed by Fig. 4?

**Technical comments**

1. **In Abstract,** remove JMA WRF-3DVAR abbreviations.

2. Please keep the abbreviation order: some are "Abbreviation (extended)" and some

extended form (abbreviation) (Lines: 163-164). Please fix this from the whole manuscript.

3. Line: 228, Model center is (17.5 °N, 140 °E) (Fig. 4). ?? please make a complete sentence.

*Editorial comments*

1. **"**the background field of the model is effectively corrected…" →"…was effectively corrected…". **Please consider whether the authors want to keep "past form of a verb. In my opinion, if the authors are explaining the results of this work, it should be the past form of a verb. (not critical)**

2. Lines: 278, 280, "Fig. 7a, c, e" (go through the whole manuscript) →"Figs. 7a, c, and e" please put this to the end of this sentence.

---

## Referee Report (RR2)

This paper built an interface for AHI radiance data assimilation on the WRFDA system based on the 3DVAR assimilation method. Two experiments for comparison was designed to examine the effect of AHI water vapor channel radiance data assimilation on the analysis and prediction of the rapid intensification period of Typhoon Soudelor in 2015. To some extent, the assimilation of AHI radiance data is able to improve the analyses of the minimum sea level pressure, the maximum surface wind, as well as the typhoon track. The whole developing stages of Typhoon Soudelor including a rapid intensification, a weakening, a second intensification, then a continuous weakening till disappearing. However, only the first intensification during 1 August to 3 August considered as study period seems insufficient to efficiently prove the advantages of AHI radiance data assimilation. According to the comparison of the two experiments during 48 hours forecasting period (Fig.13), the forecast error of AHI_DA model in the first 30 hours is obviously smaller than the CTNL model's result, however in the later 18 hours the forecast error between these two models is quite close. That means the forecasting error could possibly seriously increase for a longer simulation time. Thus in order to more efficiently prove the advantages of AHI radiance data assimilation and promote the contributions of this paper, I suggest this research to extend the study period at least include the first intensification, a weakening, and the second intensification of Typhoon Soudelor. In addition, some unclear and unprecise descriptions need carefully to be addressed. Overall speaking, this paper can be considered for publication however the major revision is necessary.

*Specific comments*

1. P. 9, Ln 169-177, please add the references for the procedures for AHI radiance data quality control.
2. P. 9, Ln 182, $N_p$ needs a definition.
3. P. 12, Ln 236, why you use 6 hours spin-up time? Please give more explanation.
4. P. 13, Ln 256-260, please add the sensitivity experiment results or some references.
5. P. 14, Ln 273, how to tell the gradient in Fig. 6b decreases stably with increasing iterations?   It keeps decreasing.
6. P. 14, Ln 273-276, The exponential decrease of the cost function and the change trend of its gradient indicate that the effectiveness of AHI radiance DA. What's the optimal value of log(gradient)?   How to see the final iterated analytical field is close to the observation?
7. P. 14, Ln 279, what is "analytical brightness temperature"?
8. P. 14, Ln 281-284, "It should be pointed that even only parts of the AHI radiance

data are applied after quality control in the data assimilation, the radiative transfer model is able to simulate the brightness temperature for all the pixels with the background and the analysis respectively for the verification purpose." This description is unprecise, at least how much should AHI radiance data be considered?

9. P. 15, Ln 309-P. 16, Ln 312, the observed and background brightness temperature for ch8 (a→b) and ch9 (d→e) both have significant improvement after the bias correction. However we can't find the similar trend for ch10, please explain.

10. P. 16, Ln 322-324, why the OMB number keeps the same with or without bias correction in Fig. 9(a)?   As well as the Stdv(K) of OMB almost keeps the same with or without bias correction in Fig. 9(c).

11. P. 17, Ln 351-352, after the assimilation of AHI radiance data, except the streamlines in the typhoon region become denser, the upper left region somehow showed quite different streamline pattern. Is this also part of improvements?

12. P. 19, Ln 379-380, "the track predicted by AHI_DA match better with the best track". This description is unprecise, only better match at the start point and the end point. There still have not small track error during the middle region.

13. P. 19, Ln 392-393, "It can be seen that the maximum surface wind error predicted by the AHI_DA is much lower than that by the CTNL..", This description is only valid before 30 hours of forecast time, but after 30 hours both models show similar error degree.

14. P. 20, Ln 395-396, "The maximum surface wind predicted by AHI_DA fit closer to the best track data with the maximum difference about 2.6 m/s after 12 hours forecast". This description seems not matching with Fig.13(a).

15. P. 21, Ln 416-418, conclusion 3 "It is found that the track, maximum surface wind, and minimum sea level pressure from the AHI radiance data assimilation experiment match better with the best track than the control experiment does for the subsequent 18-hour forecast". This conclusion doesn't match with the findings from Fig.12 and Fig.13.

16. P. 31, Fig.3, the legend is wrong. The dash line should be maximum surface wind and the solid line should be minimum sea level pressure.

17. P. 32, Fig.4, what the different symbols (triangle and circle) represent?

18. P. 38, Fig.10, please add the trend line for each channel in order for better comparison.

19. P. 40, Fig.12, the unit of track error (m s$^{-1}$) in the figure caption is wrong, it should be "km".

20. P. 41, Fig.13, the figure caption should be ….maximum surface wind "error" (unit:

m s$^{-1}$)…..minimum sea level pressure "error" (unit: hpa). In addition, the typhoon name "Soudelor" is misspelled as "Soulder".

**Technical corrections**

1. P. 1,Ln 2, the typhoon name "Soudelor" was misspells as "Soulder" in the paper title.
2. P. 6, Ln 110, "weather" forecast.
3. P. 3, Ln 57, the cited reference "Pennie, 2010" is not listed in the references.
4. P. 25, Ln 515, this reference is not cited in the article.
5. P. 26, Ln 523, this reference is not cited in the article.

---

## Author Response (AR2)

[revised manuscript text omitted]

**Reply to reviewer 1**

Major comment

1. A very critical error in title and Fig. 13. "Typhoon Soulder" →"Typhoon Soudelor", Also, please specify Soudelor's year for the clarity.
* * *
Reply: Thanks. Corrected as "Fig.13 The 48-hour (a) maximum surface wind (unit: m s$^{-1}$), (b)

minimum sea level pressure (unit: hPa) of Soudelor (2015) averaged from two forecasts."

2. Line 232: "Grell Devenyi cumulus parameterization scheme (Grell et al., 2002)," By the way, it is not always true. Nevertheless, since the authors set a 5-km spatial resolution in this study, I am very wondering why the authors did activate Grell Devenyi cumulus parameterization?
* * *
Reply: Agreed. In fact, we also conduct sensitivity experiments to turn off the cumulus parameterization for this case. It is found that there is no significant difference between them. The application of Grell Devenyi cumulus parameterization scheme we used in our current study is following that used in Li et al., (2012) with 5-km spatial resolution.

Reference:

Li, Y., Wang, X., Xue, M., 2012. Assimilation of radar radial velocity data with the WRF

    ensemble-3DVAR hybrid system for the prediction of hurricane Ike (2008). Mon. Weather

    Rev. 140, 3507–3524.

Minor comments

1. Lines 59-62: the sentence starts with "some researches …" but the authors cited one paper"

Please cite more papers or revise this sentence.
* * *
Reply: Thanks. Corrected as "Some researches demonstrated that in global model, satellite radiance DA makes more contribution to improving the accuracy of the numerical model results than conventional observation DA does (Zapotocny et al., 2007, Yan et al., 2010; Geer et al.,
2017)."
Reference added:
Geer, A. J., Baordo, F., Bormann, N., English, S., Kazumori, M., Lawrence, H., Lean, P., Lonitz,
K., and Lupu, C.: The growing impact of satellite observations sensitive to humidity, cloud
and precipitation, Quart. J. Roy. Meteorol. Soc., 143, 3189–3206.
Yan, B., F. Weng, and J. Derber (2010), Assimilation of satellite microwave water vapor sounding
channel data in NCEP Global Forecast System (GFS), paper presented at 17th International
TOVS Study Conference, Int. ATOVS Working Group, Monterrey, Calif.

2. Lines 68-69: I cannot understand what the authors mean (in bold) "it is highlighted that they are
not able to perform continuous monitoring over a fixed area, **thus leaving out some rapidly**
**intensified TCs or storms.**"
----------------------------------
Reply: Thanks. The sentence is revised as "However, it is highlighted that they are not able to
generate continuous observations for a fixed regional area and so may miss rapidly intensified TCs
or storms."

3. Line 70: "because geostationary satellites have a fixed location related to the earth's surface," it
could potentially give a misunderstanding to the reader. Please just say "rotate with the earth".
----------------------------------
Reply: Agreed. The sentence is revised as "However, it is highlighted that they are not able to
generate continuous observations for a fixed regional area and so may miss rapidly intensified TCs
or storms. On the contrary, because geostationary satellites rotate with the earth, although their
resolutions are lower than that of polar-orbit satellites, they can capture the formation and
development of mesoscale convective systems by continuous monitoring (Montmerle et al., 2007;
Stengel et al., 2009; Zou et al.,2011)."

4. Line 76: "In fact, they can capture convective spiral cloud systems relating to TCs." Since the geostationary satellites can capture more features related to TCs, the authors need to consider make a list or remove this sentence.
* * *
Reply: Agreed. The sentence is removed according to the reviewer's suggestion.

5. Lines: 236-262: is there any reason why the authors explain first Figs. 5 and 6 and followed by Fig. 4?
* * *
Reply: Figure 4 is firstly explained at line 231 before Figure 5 at line 238and Figure 6 at line 273.

Technical comments

1.  In Abstract, remove JMA WRF-3DVAR abbreviations.
* * *
Reply: Corrected.

2. Please keep the abbreviation order: some are "Abbreviation (extended)" and some extended form (abbreviation) (Lines: 163-164). Please fix this from the whole manuscript.
* * *
Reply: Agreed. All related sentence is revised as the format of extended (Abbreviation).

3.Line: 228, Model center is (17.5 °N, 140 °E) (Fig. 4). ?? please make a complete sentence.
* * *
Reply: Agreed. The sentence is completed as "The center of the model domain is located at (17.5 °N, 140 °E) (Fig. 4)."

Editorial comments

1. "the background field of the model is effectively corrected…" →"…was effectively corrected…". Please consider whether the authors want to keep "past form of a verb. In my opinion, if the authors are explaining the results of this work, it should be the past form of a verb.

(not critical)

----------------------------------

Reply: Agreed. Following the editor's suggestion, the past form is applied for the whole manuscript.

2. Lines: 278, 280, "Fig. 7a, c, e" (go through the whole manuscript) →"Figs. 7a, c, and e" .

----------------------------------

Reply:   Corrected.

**Reply to reviewer 2**

General comments

This paper built an interface for AHI radiance data assimilation on the WRFDA system based on the 3DVAR assimilation method. Two experiments for comparison was designed to examine the effect of AHI water vapor channel radiance data assimilation on the analysis and prediction of the rapid intensification period of Typhoon Soudelor in 2015. To some extent, the assimilation of AHI radiance data is able to improve the analyses of the minimum sea level pressure, the maximum surface wind, as well as the typhoon track. The whole developing stages of Typhoon Soudelor including a rapid intensification, a weakening, a second intensification, then a continuous weakening till disappearing. However, only the first intensification during 1 August to 3 August considered as study period seems insufficient to efficiently prove the advantages of AHI radiance data assimilation. According to the comparison of the two experiments during 48 hours forecasting period (Fig.13), the forecast error of AHI_DA model in the first 30 hours is obviously smaller than the CTNL model's result, however in the later 18 hours the forecast error between these two models is quite close. That means the forecasting error could possibly seriously increase for a longer simulation time. Thus in order to more efficiently prove the advantages of AHI radiance data assimilation and promote the contributions of this paper, I suggest this research to extend the study period at least include the first intensification, a weakening, and the second intensification of Typhoon Soudelor. In addition, some unclear and unprecise descriptions need carefully to be addressed. Overall speaking, this paper can be considered for publication however the major revision is necessary.
* * *
Reply: Thanks for the helpful advice. This study focuses on the impact of the assimilating AHI radiance on the initialization and prediction of a tropical cyclone case for its rapid intensification stage from 1800 UTC 1 August 2015 to 0600 UTC 4 August 2015. Similar configuration of the numerical period is also found in Honda et al., (2018) and Minamide and Zhang (2018), which only cover the first intensity stage.

However, we strongly agree that it is worth investigating the impact of AHI data assimilation on the whole period including the first intensification, a weakening, and the second intensification of

Typhoon Soudelor to fully prove the advantages of AHI radiance data assimilation. It can be seem from Figure 3 that to include the first intensification, a weakening, and the second intensification of Typhoon, the numerical period should extend another at least 4 days. The partial cycle DA

technique and other data assimilation techniques are required to update the large scale fields after several data assimilation cycles, which is beyond the scope of this study. The idea of fully investigating the impact of AHI data assimilation on the whole period including the first intensification, a weakening, and the second intensification of Typhoon Soudelor is added in the conclusion section as "Although, the whole developing stages of Typhoon Soudelor include a rapid intensification, a weakening, a second intensification, only the first intensification during 1

August to 4 August considered as the numerical period. It is worth investigating the impact of AHI

data assimilation on the whole period including the first intensification, a weakening, and the second intensification of Typhoon Soudelor to fully prove the advantages of AHI radiance data assimilation.".

References:

Honda, T., and Coauthors. 2018: Assimilating all-sky himawari-8 satellite infrared radiances: A
case of typhoon soudelor (2015). Mon. Wea. Rev., 146, 213–229.

Minamide, M., and F. Zhang, 2018: Assimilation of all-sky infrared radiances from himawari-8
and impacts of moisture and hydrometer initialization on convection-permitting tropical
cyclone prediction. Mon. Wea. Rev., 146, 3241–3258.

Specific comments

1. P. 9, Ln 169-177, please add the references for the procedures for AHI radiance data quality control.

-----------------------------------

Reply: Agreed. Added as "Apart from eliminating cloud pixels, other procedures are implemented inside the data assimilation framework for the quality control are as follows. (1) when reading the data, remove the observed outliers with values below 50 K or above 550 K; (2) only the marine observations are applied by removing the observations on the land and the observations over complex surfaces; (3) remove observations when the observation minus the background is larger than 3 times of the observation error; (4) the pixels are removed when the cloud liquid water path calculated by the background field of the numerical model is greater than or equal to 0.2 kg/m2; (5)

eliminate the data when the observation minus background is greater than 5 K. These two parameters are used for these radiances on different sensors of various satellites such as AMSU-A,

MHS, and the Advanced Microwave Scanning Radiometer 2 (AMSR2) (Wang et al., 2018, Yang et al., 2016). "

2. P. 9, Ln 182, Np needs a definition.

-----------------------------------

Reply: Np is defined in the revised manuscript.

3. P. 12, Ln 236, why you use 6 hours spin-up time? Please give more explanation.

-----------------------------------

Reply: More explanation is added as "The 6-hour spin-up period is commonly applied to initialize the typhoon or hurricane system in the data assimilation experiments, although longer spin-up period is also acceptable to introduce more model errors in the background such as 12-hour or

24-hour."

4. P. 13, Ln 256-260, please add the sensitivity experiment results or some references.

----------------------------------

Reply: Thanks. Corrected as "Also, sensitivity experiments with 25 km, and 30 km thinning mesh are also conducted with similar results (Wang et al., 2018)."

5. P. 14, Ln 273, how to tell the gradient in Fig. 6b decreases stably with increasing iterations? It keeps decreasing.

----------------------------------

Reply: Agreed, we have revised it as "Besides, the gradient in Fig. 6b decreases generally with increasing iterations."

6. P. 14, Ln 273-276, The exponential decrease of the cost function and the change trend of its gradient indicate that the effectiveness of AHI radiance DA. What's the optimal value of log(gradient)? How to see the final iterated analytical field is close to the observation?

----------------------------------

Reply: Minimization stops when the norm of the gradient for the cost function is reduced by a factor of 0.01, which is commonly used in data assimilation procedures. Inner minimization stops either when the criterion of the cost function gradient is met or when inner iterations reach 200.

Related explanation is added in section 4.1. To verify whether the final iterated analytical field is close to the observation, the observations minus the analyses are provided in section 4.2.

7. P. 14, Ln 279, what is "analytical brightness temperature"?

----------------------------------

Reply: The phase is revised as "the simulated brightness temperature from the analyses"

8. P. 14, Ln 281-284, "It should be pointed that even only parts of the AHI radiance data are applied after quality control in the data assimilation, the radiative transfer model is able to simulate the brightness temperature for all the pixels with the background and the analysis respectively for the verification purpose." This description is unprecise, at least how much should AHI radiance data be considered?
* * *
Reply: Agreed. Related information is added as "It should be pointed that even only parts of the AHI radiance data (roughly 20000 clear sky pixels of total 50000 pixels for each DA cycle) are applied after quality control in the data assimilation, the radiative transfer model is able to simulate the brightness temperature for all the model grid point with the background and the analysis respectively for the verification purpose."

9. P. 15, Ln 309-P. 16, Ln 312, the observed and background brightness temperature for ch8 (a→b) and ch9 (d→e) both have significant improvement after the bias correction. However we can't find the similar trend for ch10, please explain.
* * *
Reply: Related explanation is added as "Among them the RMSEs of channel 10 are smallest compared to those from channels 8 and 9 for the OMB and OMA samples, which is likely related to strict cloud detection scheme for channel 10 with rather lower detecting peak (Wang et al., 2018)."

10. P. 16, Ln 322-324, why the OMB number keeps the same with or without bias correction in Fig. 9(a)? As well as the Stdv (K) of OMB almost keeps the same with or without bias correction in Fig. 9(c).
* * *
Reply: Since the bias correction scheme is applied to all the AHI data after quality control, the OMB number keeps same with or without the bias correction procedure. The standard deviations (stdv) of OMB were comparable before and after the bias correction, since they are calculated by subtracting the mean of the bias. It is found that the bias was corrected effectively with an overall same magnitude of bias for each pixel, leading the stdv almost same before and after the bias correction. Related explanation is added in section 4.2 at line 348.

11. P. 17, Ln 351-352, after the assimilation of AHI radiance data, except the streamlines in the typhoon region become denser, the upper left region somehow showed quite different streamline pattern. Is this also part of improvements?

----------------------------------

Reply: The field outside of the typhoon center is also modified with the data assimilation of AHI

radiance data to provide a favorable environment field for a developing vortex. The related explanation is added as "This suggests that the field outside of the typhoon center is also adjusted as the assimilation of AHI radiance data was able to improve the large-scale environmental field in the simulation region of Typhoon Soudelor."

12. P. 19, Ln 379-380, "the track predicted by AHI_DA match better with the best track". This description is unprecise, only better match at the start point and the end point. There still have not small track error during the middle region.

----------------------------------

Reply: Agreed. The improvement is most obvious at the start and end point. From both the

48-hour predicted tracks from 0000 UTC 2 August (Fig. 12a) and from 0600 UTC 2 August (Fig.

12b), the typhoon track predicted by the CTNL continues to show a south-west bias (the red track), while the track predicted by AHI_DA (the blue track) match better with the best track (the black track). The sentences are revised as "The improvement is most obvious at the start and end point.

As can be seen in Fig. 12a, at the beginning of the forecast, the initial location of the typhoon from the CTNL experiment has large south bias and east bias at 0000 UTC and 0600 UTC respectively.

Conversely, the location of the typhoon in AHI_DA is relatively closer to the observation at the beginning."

13. P. 19, Ln 392-393, "It can be seen that the maximum surface wind error predicted by the

AHI_DA is much lower than that by the CTNL..", This description is only valid before 30 hours of forecast time, but after 30 hours both models show similar error degree.

----------------------------------

Reply: The statements are modified as "It can be seen that the maximum surface wind error predicted by the AHI_DA was much lower than that by the CTNL for the first 30 hours, due to the overall under estimation for the intensity of Typhoon Soudelor simulated in the background field.

The maximum surface wind errors of AHI_DA are generally smaller than those of CTNL. It should be pointed out that the difference between the maximum surface wind errors of the two experiments reaches up to 7.5 m s-1 after 24-hour forecast. In Fig. 13b, the results of the minimum sea level pressure are consistent with Fig. 13a, while the improvement for the minimum sea level pressure lasts for 40 hours."

14. P. 20, Ln 395-396, "The maximum surface wind predicted by AHI_DA fit closer to the best track data with the maximum difference about 2.6 m/s after 12 hours forecast". This description seems not matching with Fig.13(a).

----------------------------------

Reply: Agreed. Sorry for the typo we made. The sentence is revised as "The maximum surface wind errors of AHI_DA are generally smaller than those of CTNL. It should be pointed out that the difference between the maximum surface wind errors of the two experiments reaches up to 7.5

m s-1 after 24-hour forecast."

15. P. 21, Ln 416-418, conclusion 3 "It is found that the track, maximum surface wind, and minimum sea level pressure from the AHI radiance data assimilation experiment match better with the best track than the control experiment does for the subsequent 18-hour forecast". This conclusion doesn't match with the findings from Fig.12 and Fig.13.

----------------------------------

Reply: The sentence is corrected as "Generally, the track and minimum sea level pressure from the

AHI radiance data assimilation experiment match better with the best track than the control experiment does for the subsequent 48-hour forecast. The maximum surface wind forecast error is reduced only for the first 30-hour."

16. P. 31, Fig.3, the legend is wrong. The dash line should be maximum surface wind and the solid line should be minimum sea level pressure.

----------------------------------

Reply: Thanks. Fig. 3 is replotted.

17. P. 32, Fig.4, what the different symbols (triangle and circle) represent?

----------------------------------

Reply: Agreed. To make it clear, a sentence is added as "Each observation type is marked with different color along with a unique symbol."

18. P. 38, Fig.10, please add the trend line for each channel in order for better comparison.

----------------------------------

Reply: Agreed. The trend lines are added in the revised manuscript.

19. P. 40, Fig.12, the unit of track error (m s-1) in the figure caption is wrong, it should be "km".

----------------------------------

Reply: Thanks. The unit is corrected as km.

20. P. 41, Fig.13, the figure caption should be ….maximum surface wind "error" (unit: m s-1)…..minimum sea level pressure "error" (unit: hpa). In addition, the typhoon name "Soudelor"

is misspelled as "Soulder".

----------------------------------

Reply: The caption is revised according to the reviewer as "The 48-hour (a) maximum surface wind error (unit: m s-1), (b) minimum sea level pressure error (unit: hPa) of Soudelor (2015)

averaged from two forecasts."

Technical corrections

1.   P. 1,Ln 2, the typhoon name "Soudelor" was misspells as "Soulder" in the paper title.

--------------------------------

Reply: Corrected.

2.   P. 6, Ln 110, "weather" forecast.

--------------------------------

Reply: Corrected as weather forecast.

3.   P. 3, Ln 57, the cited reference "Pennie, 2010" is not listed in the references.

--------------------------------

Reply: The reference of Pennie, 2010 is deleted in the revised manuscript.

4.   P. 25, Ln 515, this reference is not cited in the article.

--------------------------------

Reply: The reference is deleted.

5. P. 26, Ln 523, this reference is not cited in the article.

--------------------------------

Reply: The reference is deleted.

**Reply to reviewer 3**

1. The aim of this topic is Typhoon Soudelor (2015), but the title of this paper is miss-typed.

Reply: Corrected.

2. The time notation of Fig. 3 should be more clear as mentioned at line 213.
* * *
Reply: Fig.3 is modified for the time notation for the X axis. The caption is revised to make it more clear as "Fig. 3 The time series of the minimum sea level pressure (solid line, unit: hPa) and the maximum surface wind (dash line, unit: m s-1) of Typhoon Soudelor from the CMA best-track data from 0000 UTC 30 July 2015 to 0600 UTC 12 August 2015. The specific period for the numerical results from 1800 UTC 1 August 2015 to 0000 UTC 3 August 2015 is highlighted in blue."

3. The main calculations (key features) of this paper are from line 170 to 177. The authors should provide more information about the decision of these parameters adopted. Moreover, the methodology is proposed by the authors or function of this code or processing chain.
* * *
Reply: For the quality control, we follow studies on the direct assimilation of radiance to exclude the useless brightness temperature. These parameters are used for these radiances on different sensors of various satellites such as Advance Microwave Sounding Unit-A (AMSU-A), Microwave Humidity Sounder (MHS), and the Advanced Microwave Scanning Radiometer 2 (AMSR2). These thresholds are implant to this GMI DA research to safely remove data, since these data are not able to provide useful information. The manuscript is revised as "Apart from eliminating cloud pixels, other procedures are implemented inside the data assimilation framework for the quality control are as follows. (1) when reading the data, remove the observed outliers with values below 50 K or above 550 K; (2) only the marine observations are applied by removing the observations on the land and the observations over complex surfaces; (3) remove observations when the observation minus the background is larger than 3 times of the observation error; (4) the pixels are removed when the cloud liquid water path calculated by the background field of the numerical model is greater than or equal to 0.2 kg/m2; (5) eliminate the data when the observation minus background is greater than 5 K. These two parameters are used for these radiances on different sensors of various satellites such as AMSUA, MHS, and AMSR2 (Wang et al., 2018,

Yang et al., 2016). "

4. The calculation requires iteration but the result presents forecast of typhoon data. Thus the CPU

hours should be provided.

----------------------------------

Reply: The wall clock times used by CTNL and AHI_DA for the data assimilation procedures are rather comparable with roughly 30 minutes and 40 minutes on a Linux workstation with 36

processors. It should be pointed out that computational cost of the deterministic forecast and the pre-process for gribbed GFS data are same in these two experiments. Related information is added in the revised manuscript in section 4.1.

---

## Author Response (AR3)

please prepare the study flow chart (with clear model parameter settings and comparisons) in the second or third chapter
* * *
Reply: Thanks for the advice. Done. A new flow chart has been plotted at line 654.